# SPEX: A Spectral Approach to Explainable Clustering

**Tal Argov**
Tel Aviv University
talargov1@mail.tau.ac.il

**Tal Wagner**
Tel Aviv University
talwag@tauex.tau.ac.il

## Abstract

Explainable clustering by axis-aligned decision trees was introduced by [33] and has gained considerable interest. Prior work has focused on minimizing the price of explainability for specific clustering objectives, lacking a general method to fit an explanation tree to any given clustering, without restrictions. In this work, we propose a new and generic approach to explainable clustering, based on spectral graph partitioning. With it, we design an explainable clustering algorithm that can fit an explanation tree to any given non-explainable clustering, or directly to the dataset itself. Moreover, we show that prior algorithms can also be interpreted as graph partitioning, through a generalized framework due to [45] wherein cuts are optimized in two graphs simultaneously. Our experiments show the favorable performance of our method compared to baselines on a range of datasets.

## 1 Introduction

As machine learning increasingly permeates daily life and forms the basis for consequential decision making in the real world, explaining its outputs in a manner that is interpretable to humans often becomes imperative. In a recent influential work, Moshkovitz et al. [33] proposed a model of explainability in clustering. In their model, a clustering of points in a feature space $\mathbb{R}^d$ is *explainable* if it is described by a binary decision tree, where each internal node corresponds to thresholding the points along a single coordinate. Thus, the assignment of points to clusters can be described by a sequence of individual feature thresholds, arguably making it easy to explain and interpret. See Figure 1 for illustration.

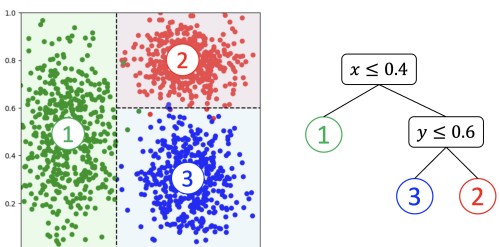

Figure 1: Illustration of explainable clustering. Clusters are generated from three gaussians. The dashed lines on the left and the decision tree on the right define the explainable clustering regions, with some points attributed to the wrong cluster.

Moshkovitz et al. [33] presented the Iterative Mistake Minimization (IMM) algorithm, which takes as input an already computed $k$-medians or $k$-means clustering of the data, called the *reference clustering*, and "rounds" into an explainable clustering. It works by fitting the reference clustering with a decision tree that greedily minimizes wrong point-to-cluster assignments. They proved that the loss in clustering cost, called the *price of explainability*, can be bounded as a function of $k$. This has led to surge of theoretical work on bounding the price of explainability, culminating in tight bounds for $k$-medians and nearly tight bounds for $k$-means [26, 28, 30, 31, 32, 12, 7, 19, 20, 4, 27].

This voluminous body of work has so far mostly focused on $k$-medians and $k$-means. These methods require the reference clustering to be endowed with centroids in order to work. This fails to capture widely used notions of clustering that do not produce centroids, like kernel $k$-means or spectral clustering [40, 10, 47], which one might wish to use as the reference clustering. [15] recently took

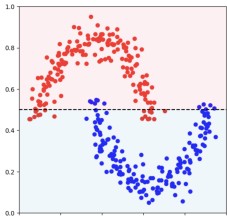 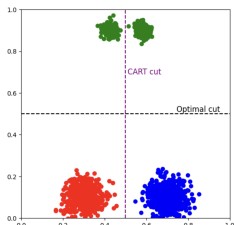

Figure 2: **Left:** The two-moons dataset admits an explainable clustering with small error. $k$-Means clustering will fail to capture the moons if used as reference clustering. Kernel $k$-means and spectral clustering capture the moons correctly, and can be rounded to the optimal explainable clustering. **Right:** A 3-way clustering example from [33]. The horizontal cut leads to an error-free explainable clustering if chosen at the first step, but CART will select the error-heavy vertical cut first.

a first step in this direction, introducing the Kernel IMM algorithm, which extends IMM to handle kernel $k$-means as the reference clustering, for certain types of kernels.

Ideally, one would want a generic explainable clustering method that can be composed over any given reference clustering, irrespective of how it was computed, whether it has centroids, or any other constraint. The only such approach at present is through repurposing classical methods for supervised classification with decision trees, which predate the definition of explainable clustering by decades. CART [6, 37] is a method for decision tree learning over a labeled dataset,[1] which outputs a tree with nodes corresponding to feature thresholds and leaves corresponding to class labels, with the aim of minimizing classification errors with respect to supervised labels. Thus, given any reference clustering, its cluster assignments can be viewed as supervised labels, and CART can be used to round it into an explainable clustering.

**Limitations of current methods.** As discussed above, explainable clustering methods that require a centroid-based reference clustering, like $k$-means, are unable to handle cluster structures not captured by centroids. This was the motivation in [15], who gave the example of the classical "two moons" dataset, depicted in Figure 2 (left). Since $k$-means fails to capture the moon structure, any explainable clustering method built on it as the reference will fail as well. On the other hand, kernel $k$-means and spectral clustering capture the moons correctly, and can be rounded to the explainable rounding depicted in the figure, with a small number of errors.

While the Kernel IMM algorithm does not require centroids, it is limited to kernel $k$-means with specific kernels, and varies by the specific kernel. Hence, it cannot be used as a generic method. Furthermore, the method is rather complex, and is evaluated in [15] on datasets of size only up to hundreds of points, leaving its scalability unclear. Our larger scale experiments indeed show that Kernel IMM becomes infeasible on bigger data.

The CART algorithm, as mentioned above, can be repurposed for explainable clustering even though it is not designed for this task. To demonstrate its drawbacks, [33] gave a toy example where CART fails to find a simple error-free explainable clustering. It is shown in Figure 2 (right), and we revisit this example in more detail in Section 3.3. There have been some questions whether this failure mode is representative of behavior on real data, or merely a pathological case not encountered in practice.[2] Preliminary small scale results in [15] implied that CART performs well empirically. However, our larger scale experiments will show that CART indeed suffers low performance on real data.

The upshot is that despite a large body of work, there is no generic method for explainable clustering, which is oblivious to the type of reference clustering and robust across datasets. This is the gap that we address in this work.

**Our results.** We introduce a new approach to explainable clustering, based on spectral graph techniques. Cheeger's inequality relates the eigenvalues of a graph to its optimal cut conductance, a fundamental graph partitioning objective that yields high-quality cuts. This useful fact is widely used

---

[1]CART is in fact a family of algorithms for Classification And Regression Trees (hence its name). The specific form of CART used in explainable clustering is detailed later, in Section 3.3. See also Section B.1.

[2]See the peer review discussion for [15] at `https://openreview.net/forum?id=FAGtjl7HOw`.

in algorithms. A geometric consequence of it, stated in Theorem 2.2, relates coordinate cuts in $\mathbb{R}^d$ to cut conductances in the graph, provided that the graph usefully captures geometric relations between the data points in $\mathbb{R}^d$. This hints at a relevance to explainable clustering.

We use this connection in two ways:

- SPEX-Clique: We may describe any given reference clustering with a graph that contains a clique over each cluster. Applying the spectral result iteratively to this graph leads to an algorithm that can round any given reference clustering, without limitations, into an explainable clustering.
- SPEX-kNN: We may also use the spectral result on a graph built directly on the dataset in $\mathbb{R}^d$, like the $k$-nearest neighbor graph. This leads to a "reference-free" algorithm, which computes an explainable clustering directly from the data, without using a reference clustering at all.

To gain more insight into existing methods and their relation to ours, we show they are captured by a generalized graph partitioning framework of non-uniform sparse cuts [45], allowing us to view them through a unified analytic lens. Namely, prior methods can be seen as different choices of graphs to describe the reference clustering.

Our experiments on a range of datasets and baseline shed light on the empirical performance of explainable clustering methods and showcases the advantage of our approach.

## 1.1 Related Work

Much work has focused on proving multiplicative bounds on the price of explainability for $k$-medians and $k$-means. In deterministic algorithms, IMM [33] achieved $O(k)$ for $k$-medians and $O(k^2)$ for $k$-means. [12] improved the $k$-means bound to $O(k \log k)$, with an algorithm we will call EMN.

Randomized methods have enabled better bounds. For $k$-medians, a sequence of works analyzed a natural randomized procedure and ultimately obtained a tight bound of $(1 + o(1)) \ln k$ [30, 32, 12, 19, 20]. For $k$-means, [20] achieved $O(k \ln \ln k)$, which is tight up to the $\ln \ln k$ term. Some works have also shown better bounds in the low-dimensional data regime [26, 12, 7], and studied the computational complexity of approximating the optimal explainable clustering [4, 20, 27].

Some works have used modified or extended definitions of tree-based explainable clustering [16, 18, 25, 41, 9, 46, 35]. In particular, [18] generalized the definition far beyond [33], allowing oblique (hyperplane) cuts that involve multiple coordinates rather than just single coordinate cuts. This greatly increases the expressive capacity of the resulting clustering—in fact, [18] show it can exactly capture any reference $k$-means clustering, without any "price of explainability"—though arguably at the price of rendering the clustering less explainable. While defining explainability in clustering remains an open-ended question, in this work we focus on the original explainability model of [33].

In the broader context of explainable machine learning, the model of [33] is an example of an *intrinsically explainability* method, where the given model (in this case, the reference clustering) is approximated by a different model (in this case, the explainable clustering tree) on which structural explainability constraints are imposed. This is in contrast with *post-hoc explainability* methods, which aim to endow the given reference model with explanations without modifying it.

## 2 Spectral Explainable Clustering

We begin with some necessary background and notation on graph partitioning. Let $G(X, E, w)$ be an undirected graph. Given a strict non-empty subset $S \subset X$, let $e_G(S, X \setminus S)$ denote the sum of edge weights crossing between $S$ and $X \setminus S$, and let $\mathrm{vol}_G(S)$ denote the sum of weighted degrees of nodes in $S$ (called the *volume* of $S$).

There are two standard notions for quantifying the outer connectivity of $S$ within $G$ (see [47]):

$$\phi_G(S) = \frac{e_G(S, X \setminus S)}{|S|} \;\; ; \;\; \psi_G(S) = \frac{e_G(S, X \setminus S)}{\mathrm{vol}_G(S)} \tag{1}$$

$\phi_G(S)$ is called the *sparsity* of $S$, while $\psi_G(S)$ is called the *conductance* of $S$. Both notions have analogues for considering $S$ and its complement $X \setminus S$ as a two-way cut:

$$\Phi_G(S) = \frac{e_G(S, X \setminus S)}{\frac{1}{|X|} \cdot |S| \cdot |X \setminus S|} \;\; ; \;\; \Psi_G(S) = \frac{e_G(S, X \setminus S)}{\frac{1}{\mathrm{vol}_G(X)} \cdot \mathrm{vol}_G(S) \cdot \mathrm{vol}_G(X \setminus S)}. \tag{2}$$

$\Phi_G(S)$ is sometimes called the *ratio cut* objective [48], while $\Psi_G(S)$ is called the *normalized cut* objective [42]. These objectives sometimes appear in the literature slightly differently, with $\min\{|S|, |X \setminus S|\}$ and $\min\{\mathrm{vol}_G(S), \mathrm{vol}_G(X \setminus S)\}$ as their respective denominators. These are the same as Equation (2) up to a factor of 2, since $\frac{1}{2}\min\{s, n-s\} \le \frac{1}{n} \cdot s \cdot (n-s) \le \min\{s, n-s\}$ for all $0 < s < n$, which applies here with $n = |X|$, $s = |S|$ in $\Phi_G(S)$, and with $n = \mathrm{vol}_G(X)$, $s = \mathrm{vol}_G(S)$ in $\Psi_G(S)$.

## 2.1 The Spectral Approach

Our approach is grounded in the following theorem, which is a generalization of a theorem given in [11], based on a framework developed by [2, 3], in the context of nearest neighbor search. It relates coordinate cuts in $\mathbb{R}^d$ to the Cheeger inequality [1, 8], a fundamental result in spectral graph theory with a myriad of algorithmic implications. While they stated the theorem specifically for the nearest neighbor graph of a point set in $\mathbb{R}^d$, it holds for general weighted graphs over points in $\mathbb{R}^d$, as we now state. We provide a proof in Section A.

**Definition 2.1.** Let $X \subset \mathbb{R}^d$. The *coordinate cut* given by coordinate $j \in \{1, \dots, d\}$ and threshold $\tau \in \mathbb{R}$ is $S_{j,\tau}(X) := \{x \in X : x_j \le \tau\}$. The cut is *valid* if $S_{j,\tau}(X) \ne \emptyset$ and $S_{j,\tau}(X) \ne X$.

**Theorem 2.2.** *Let $X \subset \mathbb{R}^d$ be a set of points, where $x \in X$ has coordinates $x = (x_1, \dots, x_d)$. Let $G(X, E, w)$ be a graph with vertex set $X$. Consider two distributions over pairs of points in $X$:*

- $\mathcal{D}_{\mathrm{adj}}$ *is the distribution over adjacent pairs in $G$, where a pair $x, y \in E$ is sampled with probability proportional to the edge weight between them.*

- $\mathcal{D}_{\mathrm{all}}$ *is the distribution over all pairs $x, y \in X$, where $x$ and $y$ are sampled independently, each with probability proportional to its weighted degree in $G$.*

*Then, there is a valid coordinate cut $j, \tau$ such that*

$$\Psi_G(S_{j,\tau}(X)) \le \sqrt{\frac{\mathbb{E}_{x,y \sim \mathcal{D}_{\mathrm{adj}}} \|x - y\|_2^2}{\mathbb{E}_{x,y \sim \mathcal{D}_{\mathrm{all}}} \|x - y\|_2^2}}.$$

Intuitively, the theorem draws a connection between combinatorial graph cuts and geometric properties of the set of points: if the graph nodes graph are associated with embeddings in $\mathbb{R}^d$, then there is a "good quality" graph cut, whose conductance is upper-bounded in terms of the squared Euclidean distances between the node embeddings. Furthermore, that cut is a coordinate cut.

The conductance bound is governed by the ratio of expected squared distances in $X$ according to two distributions over pairs of points: the numerator samples pairs of points adjacent in $G$, while the denominator samples any pair of points. Thus, the ratio (and hence the cut conductance) is smaller when $G$ captures a geometrically meaningful structure in the dataset, wherein adjacent pairs of points are expected to be nearer to each other than general pairs of points.

This implies an approach to explainable clustering. Given a reference clustering, we can describe it with a suitable graph $G$, and iteratively look for the coordinate cut with minimum conductance in each node of the decision tree. If the reference clustering is of good quality, in the sense that a pair of points are expected to be nearer to each other if they are clustered together, then Theorem 2.2 guarantees the existence of a low-conductance cut. At the same time, since the nearness of points in the same cluster is a "soft" requirement (it only needs to hold in expectation), it renders the theorem robust to various types of clustering, allowing for unconstrained cluster shapes, outliers, etc.

Another possibility that arises is to dispense with the reference clustering, and construct a graph that captures nearness/farness directly from the dataset. In Section 2.2 we describe the explainable tree construction with a general graph, and in Section 2.3 we discuss graph selection.

## 2.2 Iterative Tree Construction

Given a graph $G(X, E, w)$ over the dataset $X \subset \mathbb{R}^d$, our task is to construct an explainable clustering decision tree $T$. Let $\ell > 0$ be the desired number of leaves.

An explainable clustering tree is a decision tree in which every internal node $v$ is associated with a coordinate $j_v \in \{1, \dots, d\}$ and a threshold $\tau_v \in \mathbb{R}$, inducing the condition $x_{j_v} \le \tau_v$ given a point $x \in \mathbb{R}$. The coordinate thresholds associate a subset $X_v \subset X$ with each node $v$: the root is associated

with $X$, and every non-root $v$ with parent $u$ is associated with $X_v = S_{j_u,\tau_u}(X_u)$ if $v$ is the left child of $u$, or $X_v = X_u \setminus S_{j_u,\tau_u}(X_u)$ if it $v$ the right child of $u$.

Let $\mathcal{L}(T)$ denote the set of leaves in $T$. They induce partition of $X$ into clusters, $X = \cup_{v \in \mathcal{L}(T)} X_v$. We measure the quality of $T$ by a generalization of the normalized cut objective to multi-way partitions [47]:

$$\bar{\Psi}_G(T) = \sum_{v \in \mathcal{L}(T)} \psi_G(X_v). \qquad (3)$$

The smaller $\bar{\Psi}_G(T)$, the better the partition induced by its leaves. Note this generalizes the two-way cut objective $\Psi_G(S)$ (Equation (2)), since $\Psi_G(S) = \psi_G(S) + \psi_G(X \setminus S)$.

To build the tree, we start with $T$ containing only a root. As long as $T$ does not yet have the requisite number of leaves $\ell$, we choose the leaf $v$ to split next to greedily minimize $\bar{\Psi}_G(T)$. Splitting a leaf $v$ with a cut $S \subset X_v$ replaces the summand $\psi_G(X_v)$ in Equation (3) with $\psi_G(S) + \psi_G(X_v \setminus S)$, and thus $v$ is chosen to maximize the reduction in $\bar{\Psi}_T(G)$ its split would yield, which is

$$\psi_G(X_v) - \min_S \left( \psi_G(S) + \psi_G(X_v \setminus S) \right), \quad (4)$$

where $S$ ranges over all valid coordinate cuts. We associate $v$ with the $j_v, \tau_v$ corresponding to the minimizer $S$, split $v$ into two new leaves along this cut, and repeat. See Algorithm 1.

Note that Algorithm 1 has the flexibility to produce a tree $T$ with any desired number of leaves, regardless of the number of clusters in the reference clustering. In contrast, centroid-based methods like IMM, EMN and Kernel IMM are bound to produce the same number of leaves in $T$ as the number of centroids in the reference $k$-median or (kernel) $k$-means clustering, and increasing the number of leaves requires separate techniques [17, 31, 15]. Increasing the number of leaves is helpful for attaining a smaller price of explainability due to the increased expressivity of $T$, albeit at the expense of being less explainable due to its bigger size. We discuss this further in Appendix B. Our main evaluation will focus on producing trees with the same number of leaves as clusters in the reference clustering.

---

**Algorithm 1** SPEX

**input:** Dataset $X \subset \mathbb{R}^d$, graph $G(X, E, w)$, target number of clusters $\ell$
**output:** Decision tree $T$ where every internal node is associated with a coordinate $j$ and threshold $\tau$

BUILDTREE($X, G, \ell$):
    $T \leftarrow$ initialize a tree with a single node $v$
    $j, \tau \leftarrow \arg\min_{j,\tau}$ CUTSCORE($X, j, \tau$)
    $Q \leftarrow$ initialize a maximum priority for the tree leaves, with priorities given by LEAFSCORE
    $Q.push(v, X, j, \tau)$
    **while** $T$ has less than $\ell$ leaves **do**
        $v, X_v, j_v, \tau_v \leftarrow Q.pop()$
        Associate $v$ with the cut $j, \tau$
        Split $v$ into two new leaves $v_L, v_R$
        $X_{v_L} \leftarrow S_{j_v,\tau_v}(X_v)$
        $X_{v_L} \leftarrow X_v \setminus X_{v_L}$
        $j_L, \tau_L \leftarrow \arg\min_{j,\tau}$ CUTSCORE($X_{v_L}, j, \tau$)
        $j_R, \tau_R \leftarrow \arg\min_{j,\tau}$ CUTSCORE($X_{v_R}, j, \tau$)
        $Q.push(v_L, X_{v_L}, j_L, \tau_L)$
        $Q.push(v_R, X_{v_R}, j_R, \tau_R)$
    **return** $T$

---

CUTSCORE($X', j, \tau$):
    **if** $S_{j,\tau}(X) = \emptyset$ or $S_{j,\tau}(X) = X$ **then**
        **return** $\infty$
    **return** $\psi_G(S_{j,\tau}(X')) + \psi_G(X_v \setminus S_{j,\tau}(X'))$

---

LEAFSCORE($v, X', j, \tau$):
    **return** $\psi_G(X') -$ CUTSCORE($X', j, \tau$)

---

### 2.3 SPEX-Clique and SPEX-kNN

To capture a given reference clustering with a graph, there are several natural choices:

- *Clique graph:* points are adjacent if and only if they are in the same cluster (thus, every cluster becomes a clique).
- *Star graph:* if the reference clustering is endowed with centroids, each point can be made adjacent to its cluster centroid (thus, every cluster becomes a star).
- *Independent set (IS) graph:* points are adjacent if and only if they are *not* in the same cluster (thus, every cluster becomes an independent set). This is the complement of the clique graph. Here, of course, one would wish to maximize rather than minimize the cut objective.

The clique graph may seem like the most natural choice, and this is indeed the one we make. Perhaps surprisingly, as we will show in Section 3, when previous methods (IMM, EMN, CART) are interpreted through this graphical lens, they turn out to correspond to either the star graph or the IS graph. This directly relates to their limitations, like requiring centroids (in the case of the star graph) or failing the toy example from Figure 2 (in the case of the IS graph).

Given any reference clustering, using the clique graph as $G$ in Algorithm 1 yields the algorithm we call SPEX-Clique. We also consider an variant that uses no reference clustering, by constructing the nearest neighbor graph directly on the points in $X$. This yields the algorithm we call SPEX-kNN.

## 2.4 Computational Efficiency

SPEX as well as the baselines we consider share the following high-level structure. In each tree node $u$ with $n_u$ points, for each coordinate, they sort the points by that coordinate (time $O(n_u \log n_u)$) and perform a sweep-line procedure that iterates over the $n_u - 1$ prefix/suffix cuts by moving nodes from the suffix to the prefix one at a time. As each node is moved, the cut score is updated accordingly (let $S$ denote the time it takes to update), and eventually the cut with the overall optimal score is selected. Repeating this for all coordinates takes time $O(dn_u(\log n_u + S))$. In each tree level, the sum $\sum_u n_u$ is $n$ for SPEX and CART and $n + k \le 2n$ for IMM and EMN, therefore the time per level is $O(dn(\log n + S))$. Summing over up to $k - 1$ levels in the tree, the total time is $O(kdn(\log n + S))$.

The algorithms may differ on the time $S$ it takes update cut scores during sweep-line. In SPEX-Clique, we need not store the entire clique graph; rather, we only store point-cluster assignments ($O(n)$ memory). During sweep-line, we maintain two cluster histograms for the prefix and the suffix ($O(k)$ memory), from which the cut score can be updated in $S = O(1)$ time. In SPEX-kNN, letting $\kappa$ denote the number of neighbors per node (note that this is a different parameter than the number of clusters $k$), we can store the kNN graph in $O(n\kappa)$ memory and update cut scores in $S = O(\kappa)$ time.

## 3 Lens: Non-Uniform Sparse Cuts

While our spectral approach to explainable clustering may seem rather different from previous methods, in this section we show a generalized graph partitioning framework that captures them in a unified way. It is based on *non-uniform sparse cuts* as defined by Trevisan [45], where cuts are optimized simultaneously in two graphs that share the same set of nodes.

Trevisan [45] defined the Non-Uniform Sparsest Cut problem as follows. Let $G(X, E_G, w_G)$ and $H(X, E_H, w_H)$ be two graphs on the same set of nodes $X$. The $(G, H)$-sparsity of a cut $(S, X \setminus S)$ is defined as the (normalized) ratio of edges cut in $G$ to edges cut in $H$:

$$\Psi_{G,H}(S) = \frac{\frac{1}{\text{vol}_G(X)} \cdot e_G(S, X \setminus S)}{\frac{1}{\text{vol}_H(X)} \cdot e_H(S, X \setminus S)}.$$

The goal in the Non-Uniform Sparsest Cut problem is to find the cut with the smallest $\Psi_{G,H}(S)$.

In [45] it is observed that this generalized graph partitioning problem captures several classical problems defined on a single graph $G$ as special cases:

- The classical ("uniform") Sparsest Cut problem, of minimizing $\Phi_G(S)$ from Equation (2), is the case of $\Psi_{G,H}(S)$ where $H$ is an unweighted clique over $X$.
- The Normalized Cut problem, of minimizing $\Psi_G(S)$ from Equation (2), is the case of $\Psi_{G,H}(S)$ where $H$ is the $G$-*degree weighted clique* over $X$, in which the edge weight in $H$ between every pair $x, y \in X$ is the product of their weighted degrees in $G$.
- The Minimum $st$-Cut problem is the case of $\Psi_{G,H}(S)$ where $H$ contains a single edge between a "distinguished" pair $s, t \in X$.

Here, we further observe that this framework is useful in capturing prior methods for explainable clustering (or close variants of those methods), as they in fact produce coordinate cuts that minimize $\Psi_{G,H}(S)$ with particular choices of graphs $G$ and $H$, even though neither method is originally given in terms of graphical sparse cut terms.

- IMM [33] corresponds to $G$ being the star graph over a given reference clustering endowed with centroids, and $H$ containing a single edge that connects a pair of diametrical (furthest) centroids.
- EMN [12] also corresponds to $G$ being the star graph, but with $H$ being an unweighted clique over the $k$ cluster centroids.
- CART closely corresponds to $H$ being the independent set (IS) graph over the reference clustering (see Section 2.3), and $G$ being the $H$-degree weighted clique.

We now discuss each algorithm in turn, and highlight useful implications of this graph-theortic lens.

## 3.1 IMM as Non-Uniform Sparse Cut

Given a reference clustering $\mathcal{C} = (C_1, \ldots, C_k)$ endowed with cluster centroids $M = \{\mu^{(1)}, \ldots, \mu^{(k)}\}$, the IMM algorithm builds a decision tree in which every node $u$ is associated with a subset $X_u \subset X \cup M$ of points and centroids. In each node, it chooses the coordinate cut that minimizes the number of "mistakes", i.e., of points placed at a different side of the cut than their centroid, while separating at least one pair of centroids. Formally, for $x \in X$, let $\mu(x)$ denote the centroid of the cluster containing $x$. The cut $S \subset X_u$ in a node $u$ is chosen by IMM to minimize,

$$\mathrm{mis}_u(S) = |\{x \in S : \mu(x) \in X_u \setminus S\} \cup \{x \in X_u \setminus S : \mu(x) \in S\}|,$$

among all coordinate cuts $S$ that satisfy $\mu^{(i)} \in S$ and $\mu^{(j)} \in X \setminus S$ for at least one pair $\mu^{(i)}, \mu^{(j)} \in M \cap X_u$. The construction terminates when each leaf in the tree contains exactly one centroid.

To cast this as non-uniform sparse cut, consider a slightly modified variant of IMM. For each node $u$, Let $\mu', \mu'' \in X_u$ be a pair of centroids at maximal distance among the centroids in $M \cap X_u$. Choose the coordinate cut $S \subset X_u$ that minimizes $\mathrm{mis}_u(S)$ among those that separate $\mu', \mu''$. This is a subset of the cuts in the original IMM. For this variant, let $G$ be the *star graph* over the reference clustering, and $G_u = G[X_u]$ be its restriction to the subset $X_u \subset X \cup M$ associated with each tree node $u$. Then $\mathrm{mis}_u(S) = e_{G_u}(S, X_u \setminus S)$, since each cut edge marks a point separated from its centroid. Let $H_u$ be the *single edge graph* over $X_u$ that contains only an edge connecting $\mu'$ and $\mu''$. Then we have, $e_{H_u}(S, X_u \setminus S) = 1$ if the cut $(S, X_u \setminus S)$ separates $\mu', \mu''$, and $e_{H_u}(S, X_u \setminus S) = 0$ otherwise. Thus, minimizing $\mathrm{mis}_u(S)$ subject to a cut separating $\mu', \mu''$ is equivalent to minimizing the ratio $e_{G_u}(S, X_u \setminus S)/e_{H_u}(S, X_u \setminus S)$. This is equal (up to normalization) to the non-uniform cut sparsity, $\Psi_{G_u, H_u}(S)$. Even though this modified IMM considers less cuts than the original IMM, it attains the same price of explainability for $k$-medians, by a different proof based on graph partitioning, that avoids the intricate potential function analysis in [33]. We show this in Section A.3.

## 3.2 EMN as Non-Uniform Sparse Cut

The EMN algorithm [12] is an improvement of IMM by the following modification: in every tree node $u$, instead of minimizing $\mathrm{mis}_u(S)$, it chooses the threshold coordinate cut $S \subset X_u$ that minimizes the ratio $\mathrm{mis}_u(S)/f_u(S)$, where [12] define $f_u(S) = \min\{|S \cap M|, |(X_u \setminus S) \cap M|\}$. As with IMM, letting $G_u$ be the *star graph* over the reference clustering when restricted to $X_u$, we have $\mathrm{mis}_u(S) = e_{G_u}(S, X_u \setminus S)$. Let $H_u$ be the graph with vertex set $X_u$ whose edges form an *unweighted clique* over $M \cap X_u$ (the rest of the vertices in $X_u$ are isolated in $H_u$). Then,

$$e_{H_u}(S, X_u \setminus S) = |S \cap M| \cdot |(X_u \setminus S) \cap M|.$$

Recall from Section 2 that $\frac{1}{2} f_u(S) \leq \frac{1}{|M \cap X_u|} \cdot e_{H_u}(S, X_u \setminus S) \leq f_u(S)$. Thus, minimizing the ratio $\mathrm{mis}_u(S)/f_u(S)$ is equivalent, up to a factor of 2, to minimizing the ratio $e_{G_u}(S, X_u \setminus S)/e_{H_u}(S, X_u \setminus S)$. This is equivalent to minimizing the non-uniform cut sparsity $\Psi_{G_u, H_u}(S)$.

## 3.3 CART as Non-Uniform Sparse Cut

We first review the CART algorithm in the form used in explainable clustering. The *Gini impurity* of a distribution $(p_1, \ldots, p_k)$ over $k$ elements is defined as $\upsilon(p) = \sum_{i=1}^{k} p_i(1 - p_i)$. Given a point set $X \subset \mathbb{R}^d$, a reference clustering $\mathcal{C} = \{C_1, \ldots, C_k\}$, and a subset $S \subset X$, we may define a distribution $p_{\mathcal{C}}(S)$ over the clusters as $p_{\mathcal{C}}(S) = \left( \frac{|S \cap C_1|}{|S|}, \frac{|S \cap C_2|}{|S|}, \ldots, \frac{|S \cap C_k|}{|S|} \right)$, which corresponds to sampling a uniformly random point from $S$ and returning its cluster ID. The impurity of $S$ is defined as $\upsilon_{\mathcal{C}}(S) = \upsilon(p_{\mathcal{C}}(S))$, and the impurity of $(S, X \setminus S)$ as a two-way cut as,

$$\Upsilon_{\mathcal{C}}(S, X \setminus S) = \frac{|S|}{|X|} \cdot \upsilon_{\mathcal{C}}(S) + \frac{|X \setminus S|}{|X|} \cdot \upsilon_{\mathcal{C}}(X \setminus S).$$

The CART algorithm builds a decision tree starting with a root node associated with all of $X$. It then iteratively chooses a leaf node whose associated subset $X'$ maximizes $\mathrm{score}(X') = \upsilon_{\mathcal{C}}(X') - \min_{S \subset X'} \Upsilon_{\mathcal{C}}(S, X' \setminus S)$, where $S$ ranges over all coordinates threshold cuts, and splits that leaf across that cut. The score captures the impurity reduction gained by splitting $X'$ along its best cut. Note how this is analogous to Equation (4) in SPEX.

Table 1: Datasets. *Training set only. †Embedded with CLIP [38]. ‡Embedded with SBERT [39].

| | CIFAR-10* | 20Newsgroups* | MNIST* | Caltech 101 | Beans | Breast Cancer | Ecoli | Iris |
|---|---|---|---|---|---|---|---|---|
| # Points | 50,000 | 11,314 | 60,000 | 8677 | 13,611 | 569 | 336 | 150 |
| # Dimensions | 512† | 768‡ | 512† | 512† | 16 | 30 | 7 | 4 |
| # Classes | 10 | 20 | 10 | 101 | 7 | 2 | 8 | 3 |
| Reference | [24] | [5] | [29] | [13] | [23] | [44] | [21] | [14] |

Now, we show that a small modification to CART is equivalent to a case of non-uniform sparsest cut. Let $H$ be the *independent set* graph over the reference clustering $\mathcal{C}$ (see Section 2.3). For a subset $S \subset X$, Let $\alpha_H(S)$ denote the sum of edge weights with both endpoints in $S$. We can now interpret the impurity $\upsilon_{\mathcal{C}}(S)$ of a subset $S \subset X$ as,

$$\upsilon_{\mathcal{C}}(S) = \sum_{i=1}^{k} \frac{|S \cap C_i|}{|S|} \left(1 - \frac{|S \cap C_i|}{|S|}\right) = \frac{1}{|S|^2} \sum_{i=1}^{k} |S \cap C_i| (|S| - |S \cap C_i|) = \frac{2 \cdot \alpha_H(S)}{|S|^2}.$$

Thus, the two-way cut impurity becomes

$$\Upsilon_{\mathcal{C}}(S, X \setminus S) = \frac{2}{|X|} \left(\frac{\alpha_H(S)}{|S|} + \frac{\alpha_H(X \setminus S)}{|X \setminus S|}\right). \tag{5}$$

To proceed, we make the modification of replacing set cardinalities with their volumes in $H$, as in the transition from sparsity to conductance in Equations (1) and (2). We get the modified two-way cut impurity, instead of Equation (5):

$$\widetilde{\Upsilon}_{\mathcal{C}}(S, X \setminus S) = \frac{2}{\mathrm{vol}_H(X)} \left(\frac{\alpha_H(S)}{\mathrm{vol}_H(S)} + \frac{\alpha_H(X \setminus S)}{\mathrm{vol}_H(X \setminus S)}\right).$$

This is precisely (up to normalization terms) the $Nassoc$ graph cut objective defined in [42, eq. (3)].[3] As they observed, it is directly related to the normalized cut objective, i.e., to the cut conductance:

$$\Psi_H(S) = 2 - \frac{\mathrm{vol}_H(X)}{2} \cdot \widetilde{\Upsilon}_{\mathcal{C}}(S, X \setminus S).$$

Thus, finding a coordinate cut $S$ that minimizes our modified impurity $\widetilde{\Upsilon}$ is equivalent to *maximizing* the usual cut conductance $\Psi_H(S)$ in $H$. This, in turn, is equivalent to maximizing the non-uniform cut sparsity $\Psi_{H,G}(S)$, where $H$ is the *independent set* graph and $G$ is the $H$-*degree weighted clique* over $X$, since (as observed in [45] and mentioned in the beginning of this section), for this choice of graphs it holds that $\Psi_{H,G}(S) = \Psi_H(S)$, up to normalization. Finally, since $\Psi_{G,H}(S) = 1/\Psi_{H,G}(S)$ by definition, this is equivalent *minimizing* the non-uniform cut sparsity $\Psi_{H,G}(S)$.

This point of view clarifies the failure of CART in Figure 2. The IS graph over a reference clustering incentivizes cuts that separate pairs of points residing in different clusters, but has no incentive to *not* separate points residing in the same cluster, since they are not connected with any edges and hence have no effect on the cut value. Since CART maximizes cut conductance, it opts to cut the larger number of edges between the bottom clusters (vertical cut) over the smaller number of edges between the top cluster and each of the bottom clusters (horizontal cut), which is incompatible with clustering.

## 4 Experimental Evaluation

We evaluate our methods compared to baselines, on eight public real-world datasets of various sizes and dimensions, detailed in Table 1. Our code is available online.[4]

*Our algorithms.* We focus on two instantiations of our method: SPEX-Clique with spectral clustering as the reference, and SPEX-kNNwhere the $k$-NN graph is constructed with $k = 20$. Section B includes additional results for SPEX-Clique with $k$-means and kernel $k$-means as the reference clustering, and for SPEX-kNN with other values of $k$.

*Baselines.* We evaluate the following baselines:

---

[3] In the notation of [42], $\alpha_H(S) = assoc(S, S)$ and $\mathrm{vol}_H(S) = assoc(S, X)$.
[4] https://github.com/talargv/SpEx

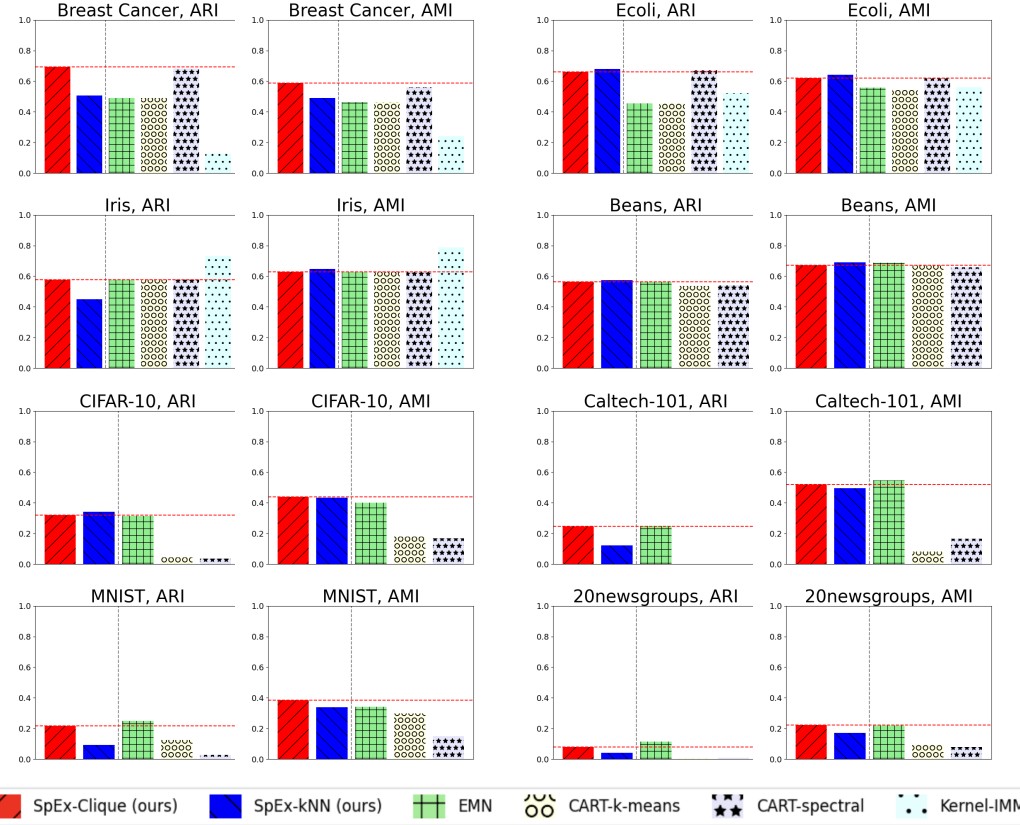

Figure 3: Adjusted Rand Index (ARI) and Adjusted Mutual Information (AMI), higher is better.

- EMN [12], the state of the art for explainable clustering with $k$-means as the reference clustering. Note that EMN cannot work with a spectral or a kernel $k$-means reference clustering, as its operation requires the reference clustering to have centroids.

- CART, with both $k$-means and spectral clustering as the reference clustering.

- Kernel IMM [15], which uses kernel $k$-means as the reference clustering.

Our experiments indicated that Kernel IMM does not scale well, failing to run on the larger scale datasets. We therefore report its results only for the three smaller scale datasets (Breast Cancer, Ecoli, Iris), which are the ones used in [15]. See more in Section B.

*Evaluation measures.* The classes of each dataset are used as groundtruth clusters. To evaluate each method, we report two standard (and incomparable) measures of clustering agreement between two clusterings of the same data: Adjusted Rand Index (ARI) [22] and Adjusted Mutual Information (AMI) [34]. Each is explainable clustering is evaluated for its agreement with the groundtruth clustering through these measures.

## 4.1 Results

Main results are reported in Figure 3. Tables 2 and 3 augment them with additional choices of reference clustering for SPEX-Clique and CART and list ARIs with respect to the reference clustering. They also contain two synthetic small-scale datasets, R15 and Pathbased, used in [15]. Appendix B includes additional results.

The results show that SPEX-Clique is the most consistently high-performing algorithm. It outperforms each baseline in most cases in both evaluation measures, and is never exceeded by more than one baseline at the same time. SPEX-kNN is highly effective particularly on the low dimensional datasets

Table 2: Results on smaller datasets. Rows are grouped by reference clustering, shown as the first row in each group. The REF column lists the ARI with respect to the reference clustering (rather than the ground truth as in ARI column). Best scores per reference clustering are in bold.

| | R15 | | | Pathbased | | | Ecoli | | | Iris | | | Cancer | | |
|---|---|---|---|---|---|---|---|---|---|---|---|---|---|---|---|
| Algorithm | ARI | AMI | REF | ARI | AMI | REF | ARI | AMI | REF | ARI | AMI | REF | ARI | AMI | REF |
| *REF:* Spectral | .993 | .994 | 1. | .526 | .570 | 1. | .711 | .653 | 1. | .630 | .661 | 1. | .743 | .626 | 1. |
| SPEX-kNN | .982 | .987 | .989 | .332 | .410 | .551 | **.679** | **.642** | .863 | .450 | **.647** | .450 | .507 | .490 | .562 |
| SPEX-Clique | **.986** | **.989** | **.993** | **.441** | **.517** | **.824** | .662 | .621 | .847 | **.576** | .629 | **.787** | **.694** | **.588** | .785 |
| CART | **.986** | **.989** | **.993** | **.441** | **.517** | **.824** | .672 | .618 | **.886** | **.576** | .629 | **.787** | 683 | .560 | **.811** |
| *REF:* $k$-means | .993 | .994 | 1. | .461 | .543 | 1. | .489 | .609 | 1. | .641 | .669 | 1. | .491 | .464 | 1. |
| EMN | **.986** | **.989** | **.993** | **.461** | **.543** | **1.** | **.456** | **.559** | .873 | **.576** | **.629** | .772 | **.491** | **.464** | **1.** |
| SPEX-Clique | **.986** | **.989** | **.993** | **.461** | **.543** | **1.** | **.456** | **.559** | **.873** | **.576** | **.629** | **.772** | **.491** | **.464** | **1.** |
| CART | **.986** | **.989** | **.993** | .421 | .507 | .897 | .454 | .543 | .840 | **.576** | **.629** | **.772** | **.491** | **.464** | **1.** |
| *REF:* Kernel $k$-means | .908 | .967 | 1. | .919 | .888 | 1. | .538 | .612 | 1. | .731 | .767 | 1. | .116 | .228 | 1. |
| Kernel IMM | **.904** | **.962** | **.986** | **.614** | **.614** | **.583** | .522 | .560 | .848 | **.732** | **.788** | **.924** | .127 | .241 | **.93** |
| SPEX-Clique | .869 | .941 | .951 | .479 | .553 | .450 | **.529** | **.573** | **.851** | **.732** | **.788** | **.924** | **.406** | **.414** | .511 |
| CART | .682 | .876 | .759 | .479 | .553 | .450 | .500 | .558 | .824 | **.732** | **.788** | **.924** | **.406** | **.414** | .511 |

Table 3: Results on larger datasets.

| | MNIST | | | Caltech 101 | | | Newsgroups | | | Beans | | | Cifar | | |
|---|---|---|---|---|---|---|---|---|---|---|---|---|---|---|---|
| Algorithm | ARI | AMI | REF | ARI | AMI | REF | ARI | AMI | REF | ARI | AMI | REF | ARI | AMI | REF |
| *REF:* Spectral | .745 | .820 | 1. | .563 | .859 | 1. | .431 | .671 | 1. | .586 | .677 | 1. | .712 | .801 | 1. |
| SPEX-kNN | .092 | .338 | .150 | .121 | .497 | .168 | .042 | .170 | .09 | **.574** | **.690** | .649 | **.342** | .434 | .373 |
| SPEX-Clique | **.217** | **.384** | **.282** | **.247** | **.521** | **.303** | **.078** | **.223** | **.189** | .564 | .671 | **.743** | .320 | **.438** | **.394** |
| CART | .027 | .148 | .030 | -.010 | .166 | .005 | .008 | .078 | -.013 | .542 | .658 | .705 | .036 | .169 | .035 |
| *REF:* $k$-means | .364 | .481 | 1. | .405 | .822 | 1. | .502 | .660 | 1. | .572 | .689 | 1. | .636 | .738 | 1. |
| EMN | **.25** | **.342** | **.42** | **.249** | **.548** | **.416** | **.115** | .219 | **.163** | **.563** | **.688** | **.780** | **.314** | .402 | **.387** |
| SPEX-Clique | .209 | .336 | .403 | .122 | .495 | .195 | .098 | **.249** | .128 | .562 | .687 | .773 | .288 | **.410** | .370 |
| CART | .124 | .299 | .229 | -.017 | .082 | .003 | .005 | .092 | .006 | .536 | .669 | .757 | .045 | .180 | .037 |

(Beans, Ecoli and Iris).[5] On these datasets, it outperforms all baselines as well as SPEX-Clique, with the exception of Kernel IMM on Iris.[6]

CART performs well on the smaller datasets (confirming similar results reported in [15]), however, it performs poorly on the larger datasets. Recall that CART is not originally intended for explainable clustering, but can be repurposed for it since it produces a structurally compatible output (a binary decision tree with coordinate cuts). In Sections 1 and 3.3, we discussed a toy example from [33] that demonstrates a failure mode of CART for explainable clustering due to its incompatible objective. Our experimental results indicate this incompatibility may also impede its performance on real data.

**Conclusion and limitations.** Prior work on explainable clustering has focused on a few specific reference objectives, allowing it to prove worst-case approximation bounds tailored to them (cf. Section 1.1). SPEX, on the other hand, fills the gap of a generic, reference-oblivious method, currently lacking in the literature. Thus, while theoretically well-grounded (Theorem 2.2, Section 3), it cannot offer such bounds. Other methods may be better choices for some specific reference objectives – e.g., EMN is known to be near-optimal for $k$-means, and Tables 2 and 3 indeed show that it is generally better than SPEX for that reference objective. SPEX is better for data where objectives already well-studied in explainable clustering (like $k$-means) fall short, and more versatile objectives (like spectral clustering) are needed. As future work, it would be interesting to explore if the theoretical framework we develop can yet lead to formal approximation bounds for some of these objectives.

---

[5]These datasets are of dimension up to 16. The other datasets have dimension at least 30.

[6]On Iris, kernel $k$-means significantly outperforms spectral and $k$-means as the reference clustering. In Figure 3, only Kernel IMM uses it a reference. When SPEX-Clique or CART are run on the same kernel $k$-means reference, the results they yield are identical to Kernel IMM. These results are included in Section B.

## Acknowledgements

This work was supported by Len Blavatnik and the Blavatnik Family foundation and by an Alon Scholarship of the Israeli Council for Higher Education. TW is also with Amazon; this work is not associated with Amazon.

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

## Impact Statement

This paper is concerned with fitting interpretable explanations to decisions made by machine learning algorithms, in this case, grouping of individual points into clusters. While explainability in AI is intended to promote safe and ethical use in machine learning, any deployment of automated systems in settings consequential to human individuals may have unforeseen consequences. While our work is not associated with any particular outstanding or new risks, every real-world use of an explainability mechanism in AI should be used with caution and incorporate human oversight, testing, and other use case specific safeguards to mitigate any associated risks.

## A   Proofs

In Section A.1, we provide a proof of Theorem 2.2. In Section A.2, as a concrete demonstration, we show an application of it that directly relates the $k$-means reference clustering cost to the graph conductance in a certain graph that describes the reference clustering. In Section A.3, we provide the modified IMM analysis discussed in Section 3.1.

### A.1   Proof of Theorem 2.2

We restate the theorem:

**Theorem A.1** (Theorem 2.2, restated). *Let $X \subset \mathbb{R}^d$ be a set of points, where $x \in X$ has coordinates $x = (x_1, \ldots, x_d)$. Let $G(X, E, w)$ be a graph with vertex set $X$. Consider the following two distributions over pairs in of points in $X$:*

- *$\mathcal{D}_{\mathrm{adj}}$ is the distribution over adjacent pairs in $G$, where a pair $x, y \in E$ is sampled with probability proportional to the edge weight between them.*

- *$\mathcal{D}_{\mathrm{all}}$ is the distribution over all pairs $x, y \in X$, where $x$ and $y$ are sampled independently, each with probability proportional to its weighted degree in $G$.*

*Then, there is a valid coordinate cut $j, \tau$ such that*

$$\Psi_G(S_{j,\tau}(X)) \leq \sqrt{\frac{\mathbb{E}_{x,y\sim\mathcal{D}_{\mathrm{adj}}}\|x-y\|_2^2}{\mathbb{E}_{x,y\sim\mathcal{D}_{\mathrm{all}}}\|x-y\|_2^2}}.$$

The proof uses arguments that are standard in spectral graph theory, and follows [11] in a generalized form (with an arbitrary graph). We include it here for completeness and clarity.

Recall the following standard useful fact,

**Fact A.2.** *Let $a_1, \ldots, a_n \geq 0$ and $b_1, \ldots, b_n \geq 0$ such that $\sum_i b_i > 0$. Then*

$$\min_i \frac{a_i}{b_i} \leq \frac{\sum_i a_i}{\sum_i b_i} \leq \max_i \frac{a_i}{b_i}.$$

Using it we have,

$$
\begin{aligned}
\frac{\mathbb{E}_{x,y\sim\mathcal{D}_{\mathrm{adj}}}\|x-y\|_2^2}{\mathbb{E}_{x,y\sim\mathcal{D}_{\mathrm{all}}}\|x-y\|_2^2} &= \frac{\mathbb{E}_{x,y\sim\mathcal{D}_{\mathrm{adj}}}\sum_{i=1}^d |x_i-y_i|^2}{\mathbb{E}_{x,y\sim\mathcal{D}_{\mathrm{all}}}\sum_{i=1}^d |x_i-y_i|^2} \\
&= \frac{\sum_{i=1}^d \mathbb{E}_{x,y\sim\mathcal{D}_{\mathrm{adj}}}|x_i-y_i|^2}{\sum_{i=1}^d \mathbb{E}_{x,y\sim\mathcal{D}_{\mathrm{all}}}|x_i-y_i|^2} \\
&\geq \min_{i^*\in\{1,\ldots,d\}} \frac{\mathbb{E}_{x,y\sim\mathcal{D}_{\mathrm{adj}}}|x_{i^*}-y_{i^*}|^2}{\mathbb{E}_{x,y\sim\mathcal{D}_{\mathrm{all}}}|x_{i^*}-y_{i^*}|^2}. \quad (6)
\end{aligned}
$$

Let $n = |X|$ be the number of nodes in $G$. For convenience we arbitrarily label them $1, \ldots, n$ so we can index the entries a vector $z \in \mathbb{R}^n$ by points in $X$, so the coordinates of $z$ are $(z(x))_{x\in X}$. We do so similarly for matrices in $\mathbb{R}^{n\times n}$.

Recall that $w(x, y)$ is the edge weight between $x$ and $y$ in $G$. Let $d_G(x)$ denote the weighted degree of $x$ in $G$. Let $\Delta_G = \sum_{x,y} w(x, y)$ be the sum of all edge weights. Let $A_G \in \mathbb{R}^{n\times n}$ be its

weighted adjacency matrix, $A_G(x,y) = w(x,y)$. Let $D_G \in \mathbb{R}^{n \times n}$ be its diagonal matrix of degrees, $D_G(x,x) = d_G(x)$. Let $L_G = D_G - A_G$ be its Laplacian matrix. A standard fact in spectral graph theory is the identity $z^T L_G z = \sum_{x,y} w(x,y)|z(x) - z(y)|^2$ for every $z \in \mathbb{R}^n$.

Let $z_* \in \mathbb{R}^n$ be the vector with entries defined by $z_*(x) = x_{i^*}$, where $i^* \in \{1, \ldots, d\}$ is the minimizer from Equation (6). Recalling that $\mathcal{D}_{\mathrm{adj}}$ is the distribution over pairs $x, y$ with probability mass $\frac{w(x,y)}{\Delta_G}$, we have

$$\mathbb{E}_{x,y \sim \mathcal{D}_{\mathrm{adj}}} |x_{i^*} - y_{i^*}|^2 = \sum_{x,y} \frac{w(x,y)}{\Delta_G} |x_{i^*} - y_{i^*}|^2$$

$$= \sum_{x,y} \frac{w(x,y)}{\Delta_G} |z_*(x) - z_*(y)|^2$$

$$= \frac{z_*^T L_G z_*}{\Delta_G}. \tag{7}$$

Let $H$ be a weighted clique over $X$ in which the edge weight between every pair $x, y \in X$ is

$$w_H(x,y) = \frac{d_G(x) \cdot d_G(y)}{2\Delta_G}. \tag{8}$$

This is the $G$-degree weighted clique over $X$ mentioned in Section 3, except we scale all weights down by $2\Delta_G$. The weighted degree of $x$ in $H$ is

$$d_H(x) = \sum_{y \neq x} w_H(x,y)$$

$$= \frac{d_G(x)}{\Delta_G} \sum_{y \neq x} d_G(y)$$

$$= \frac{d_G(x)}{2\Delta_G}(2\Delta_G - d_G(x)) \qquad \text{since } \Delta_G = \sum_{x,y} w(x,y) = \frac{1}{2}\sum_{y \in X} d_G(y)$$

$$= d_G(x) - \frac{(d_G(x))^2}{2\Delta_G}. \tag{9}$$

Let $A_H$, $D_H$ and $L_H = D_H - A_H$ be the weighted adjacency matrix of $H$, its diagonal degree matrix, and its Laplacian matrix, respectively. Let $v_G \in \mathbb{R}^n$ be the vector of weighted degrees in $G$, scaled down by $\sqrt{2\Delta_G}$:

$$v_G(x) = \frac{d_G(x)}{\sqrt{2\Delta_G}}.$$

Observe that, by Equations (8) and (9),

$$D_H(x,y) = \begin{cases} d_G(x) - \frac{(d_G(x))^2}{2\Delta_G} & \text{if } x = y \\ 0 & \text{if } x \neq y \end{cases} \quad \text{and} \quad A_H(x,y) = \begin{cases} 0 & \text{if } x = y \\ \frac{d_G(x) \cdot d_H(y)}{2\Delta_G} & \text{if } x \neq y \end{cases}$$

Therefore,

$$L_H = D_H - A_H = D_G - v_G v_G^T. \tag{10}$$

Now, recall that $\mathcal{D}_{\mathrm{all}}$ is the distribution over pairs $x, y$, where $x$ and $y$ are i.i.d., each with probability proportional to its degree in $G$. Hence,

$$\mathbb{E}_{x,y \sim \mathcal{D}_{\mathrm{all}}} |x_{i^*} - y_{i^*}|^2 = \sum_{x,y} \frac{d_G(x)}{2\Delta_G} \cdot \frac{d_G(y)}{2\Delta_G} \cdot |x_{i^*} - y_{i^*}|^2$$

$$= \frac{1}{2\Delta_G} \sum_{x,y} w_H(x,y) |z_*(x) - z_*(y)|^2 \qquad \text{by Equation (8)}$$

$$= \frac{z_*^T L_H z_*}{2\Delta_G}. \tag{11}$$

Plugging Equations (7) and (11) into Equation (6), we have obtained,

$$\frac{\mathbb{E}_{x,y\sim\mathcal{D}_{\text{adj}}}\|x-y\|_2^2}{\mathbb{E}_{x,y\sim\mathcal{D}_{\text{all}}}\|x-y\|_2^2} \geq \frac{2\cdot z_*^T L_G z_*}{z_*^T L_H z_*}. \tag{12}$$

Finally, let $\mathbf{0}$ and $\mathbf{1}$ denote the all-0 and all-1 vectors in $\mathbb{R}^n$. Let $\gamma_G$ be the constant $\gamma_G = v_G^T z_*/v_G^T \mathbf{1}$. Letting

$$\tilde{z} = z_* - \gamma_G \mathbf{1},$$

we have

$$v_G^T \tilde{z} = v_G^T(z_* - \gamma_G \mathbf{1}) = v_G^T z_* - \frac{v_G^T z_*}{v_G^T \mathbf{1}} \cdot v_G^T \mathbf{1} = 0. \tag{13}$$

It is a standard fact that every graph Laplacian $L$ satisfies $L\mathbf{1} = \mathbf{0}$. Therefore,

$$L_G z_* = L_G(\tilde{z} + \gamma_G \mathbf{1}) = L_G \tilde{z},$$

and, using Equations (10) and (13),

$$L_H z_* = L_H(\tilde{z} + \gamma_G \mathbf{1}) = L_H \tilde{z} = (D_G - v_G v_G^T)\tilde{z} = D_G \tilde{z}.$$

Plugging these into Equation (12), we obtain

$$\frac{\mathbb{E}_{x,y\sim\mathcal{D}_{\text{adj}}}\|x-y\|_2^2}{\mathbb{E}_{x,y\sim\mathcal{D}_{\text{all}}}\|x-y\|_2^2} \geq 2\cdot\frac{\tilde{z}^T L_G \tilde{z}}{\tilde{z}^T D_G \tilde{z}}. \tag{14}$$

Now we can state the Cheeger inequality for graphs, or rather, a somewhat more general and useful form of it (see, e.g., [43, Theorem 21.1.3]):

**Theorem A.3.** *Let $G(X, E, w)$ be an undirected weighted graph with $n$ nodes. Let $A_G$ be its weighted adjacency matrix, $v_G$ the vector weighted degrees, $D_G$ the diagonal matrix of weighted degrees, and $L_G = D_G - A_G$ the Laplacian.*

*Suppose we have a vector $\tilde{z} \in \mathbb{R}^n$ that satisfies $\tilde{z}^T v_G = 0$. Then, there is a threshold $\tilde{\tau} \in \mathbb{R}$ such that if we consider the cut $S = \{x \in X : \tilde{z}(x) \leq \tilde{\tau}\}$, it satisfies $S \neq \emptyset$, $S \neq X$, and*

$$\Psi_G(S) \leq \sqrt{2\cdot\frac{\tilde{z}^T L_G \tilde{z}}{\tilde{z}^T D_G \tilde{z}}}. \tag{15}$$

Theorem A.3 together with Equation (14) imply a cut with conductance

$$\Psi_G(S) \leq \sqrt{\frac{\mathbb{E}_{x,y\sim\mathcal{D}_{\text{adj}}}\|x-y\|_2^2}{\mathbb{E}_{x,y\sim\mathcal{D}_{\text{all}}}\|x-y\|_2^2}},$$

given by thresholding the coordinates of $\tilde{z}$ at $\tilde{\tau}$. Recalling that for every $x \in X$ it holds that

$$\tilde{z}(x) = z_*(x) - \gamma_G = x_{i^*} - \gamma_G,$$

the cut $S$ can be equivalently described as $S = \{x \in X : x_{i^*} \leq \tilde{\tau} + \gamma_G\}$. Therefore, it is a coordinate cut with coordinate $i^*$ and threshold $\tau = \tilde{\tau} + \gamma_G$, and Theorem 2.2 is proven. $\qquad\square$

**Remark A.4.** *For context, we relate Theorem A.3 to the more familiar form of Cheeger's inequality. Let $\hat{L}_G = D^{-1/2} L_G D^{-1/2}$ be the normalized Laplacian matrix of $G$. It is positive semi-definite and its smallest eigenvalue is 0. Let $\lambda_G$ be its second smallest eigenvalue, and $z_G$ a corresponding eigenvector. It can be shown that the quotient at right-hand side of Equation (15) is minimized by choosing $\tilde{z} = D^{1/2} z_G$, and that the minimal value it takes is $\lambda_G$. Hence the more standard form of Cheeger's inequality, $\Psi_G(S) \leq \sqrt{2\lambda_G}$.*

## A.2  $k$-Means Example of Theorem 2.2

As a demonstrative example, let us show a special cases of Theorem 2.2, in which a reference $k$-means clustering is described by a suitable graph, and observe the conductance bounds it yields. Let $\mathcal{C} = \{C_1, \ldots, C_k\}$ be a partition of $X$ into $k$ clusters with corresponding centroids $\mu^{(1)}, \ldots, \mu^{(k)}$.

In $k$-means, the centroids are given by cluster means, $\mu^{(i)} = \frac{1}{|C_i|} \sum_{x \in C_i} x$. Denote the cost of cluster $C_i$ by $\text{cost}(C_i) = \sum_{x \in C_i} \|x - \mu^{(i)}\|_2^2$, and the total reference clustering cost by

$$\text{cost}(\mathcal{C}) = \sum_{i=1}^{k} \text{cost}(C_i) = \sum_{i=1}^{k} \sum_{x \in C_i} \|x - \mu^{(i)}\|_2^2,$$

We instantiate Theorem 2.2 with a clique graph with edge weights inversely proportional to their cluster sizes.

**Corollary A.5.** *In the weighted clique graph where every edge connecting $x, y \in C_i$ has weight $1/(|C_i| - 1)$, there is a coordinate cut with conductance at most*

$$\sqrt{2 \cdot \frac{\text{cost}(\mathcal{C})}{\frac{1}{|X|} \sum_{x,y \in X} \|x - y\|_2^2}}.$$

The proof uses the following fact which can be verified by direct substitution.

**Fact A.6.** *For every set of points $C$ with mean $\mu = \frac{1}{|C|} \sum_{x \in C} x$,*

$$\sum_{x \in C} \|x - \mu\|_2^2 = \frac{1}{|C|} \sum_{x \neq y \in C} \|x - y\|_2^2.$$

*Proof.* Denote the weighted clique graph by $G(X, X^2, w)$. For this graph:

$$w(x, y) = \begin{cases} \frac{1}{|C_i| - 1} & \exists\, i\ x, y \in C_i \\ 0 & else \end{cases}$$

(Note that if $|C_i| = 1$ then the only node in $C_i$ has no incident edges, thus the above setting of edge weights is well-defined.) Thus, it holds that:

$$\forall i, x \in C_i \;\; ; \;\; d_G(x) = \sum_{\substack{y \in C_i \\ y \neq x}} \frac{1}{|C_i| - 1} = 1.$$

Hence, Theorem 2.2 implies the existence of a coordinate cut for which the conductance is at most

$$\sqrt{\frac{\mathbb{E}_{x,y \sim \mathcal{D}_{adj}} \|x - y\|_2^2}{\mathbb{E}_{x,y \sim \mathcal{D}_{all}} \|x - y\|_2^2}} = \sqrt{\frac{\sum_{x,y \in X} \frac{w(x,y)}{\sum_{x',y' \in X} w(x',y')} \|x - y\|_2^2}{\sum_{x,y \in X} \frac{d_G(x)d_G(y)}{\sum_{x',y' \in X} d_G(x)d_G(y)} \|x - y\|_2^2}}$$

$$= \sqrt{\frac{\frac{1}{|X|} \sum_{i=1}^{k} \frac{1}{|C_i| - 1} \sum_{\substack{x \neq y \\ x,y \in C_i}} \|x - y\|_2^2}{\frac{1}{|X|^2} \sum_{x,y \in X} \|x - y\|_2^2}}$$

$$\underset{\textit{Theorem A.6}}{=} \sqrt{\frac{\sum_{i=1}^{k} \frac{1}{|C_i| - 1} |C_i| \sum_{x \in C_i} \|x - \mu^{(i)}\|_2^2}{\frac{1}{|X|} \sum_{x,y \in X} \|x - y\|_2^2}}$$

$$\leq \sqrt{\frac{2 \cdot \sum_{i=1}^{k} \sum_{x \in C_i} \|x - \mu^{(i)}\|_2^2}{\frac{1}{|X|} \sum_{x,y \in X} \|x - y\|_2^2}}$$

$$= \sqrt{2 \cdot \frac{\text{cost}(\mathcal{C})}{\frac{1}{|X|} \sum_{x,y \in X} \|x - y\|_2^2}}.$$

$\square$

Note that the denominator term $\frac{1}{|X|} \sum_{x,y \in X} \|x - y\|_2^2$ is a fixed size for the dataset and is independent of the reference clustering $\mathcal{C}$. Thus, with an appropriate choice of graph to describe the reference clustering, Theorem 2.2 yields coordinate cuts with conductance directly related to the reference clustering cost.

## A.3 IMM Alternative Analysis

In this section we focus on $k$-medians clustering in the $\ell_1$ norm. For every subset $C \subset \mathbb{R}^d$, denote its median by

$$\text{med}(C) = \min_{z \in \mathbb{R}^d} \sum_{x \in C} \|x - y\|_1.$$

Let $\mathcal{C} = C_1, \ldots, C_k$ be a reference $k$-medians clustering of $X$ with respective cluster centroids $\mu^{(i)} = \text{med}(C_i)$. The $k$-medians clustering cost of $\mathcal{C}$ is defined as

$$\text{cost}_1(\mathcal{C}) = \sum_{i=1}^{k} \sum_{x \in C_i} \|x - \mu^{(i)}\|_1.$$

Let $T_{\text{IMM}}(\mathcal{C})$ be the explainable clustering tree returned by IMM algorithm. [33] proved the following theorem:

**Theorem A.7.** *For every reference clustering $\mathcal{C}$ with centroids,* $\text{cost}_1(T_{\text{IMM}}(\mathcal{C})) \leq O(k) \cdot \text{cost}_1(\mathcal{C})$.

In the terminology from Section 1, the "price of explainability" of $k$-medians clustering is $O(k)$. This remains the best bound to date for a deterministic algorithm.[7]

In this section we prove the same theorem for our slightly modified variant from Section 3.1, where it was cast in terms of minimizing the non-uniform cut sparsity. The goal is demonstrate how the graph partitioning framework can be used analytically, and present an alternative and arguably simpler proof for the same price of explainability upper bound. Let $\tilde{T}_{\text{IMM}}(\mathcal{C})$ denote the tree returned by our modified IMM.

**Theorem A.8.** *For every reference clustering $\mathcal{C}$ with centroids,* $\text{cost}_1(\tilde{T}_{\text{IMM}}(\mathcal{C})) \leq O(k) \cdot \text{cost}_1(\mathcal{C})$.

For the proof, we introduce some notation aligned with [33] for better comparison and readability. Recall that the IMM algorithm builds an explainable decision tree on both the points $X$ and the centroids $M = \{\mu^{(1)}, \ldots, \mu^{(k)}\}$, such that each tree leaf contains exactly one centroid, and partition of $X$ into the $k$ leaves forms the explainable clustering. For every $x \in X$, denote by $\mu(x) \in M$ its centroid in the reference clustering $\mathcal{C}$. For every tree node $u$, let $X^u \subset X$ and $M^u \subset M$ be the subsets of points and centroids, respectively, contained in $u$, and let $Y^u = X^u \cup M^u$. A *mistake* in $u$ is a pair $x \in X^u$, $c(x) \in M^u$ that are separated by the threshold coordinate cut in $u$. Let $t^u$ denote the number of mistakes in $u$. Also, recall the in our modified IMM, in each node $u$ we fix a pair of centroids $\mu'_u, \mu''_u$ at maximal distance among the centroids in $u$:

$$\|\mu'_u - \mu''_u\|_1 = \min_{i,j:\tilde{\mu}', \tilde{\mu}'' \in M^u} \|\tilde{\mu}' - \tilde{\mu}''\|_1.$$

The proof of Theorem A.8 has two steps, analogous to the proof of Theorem A.7. The first step is a straightforward consequence of the triangle inequality. The detailed proof appears in [33] and is omitted here.

**Lemma A.9** (Lemma 5.5 in [33])**.**

$$\text{cost}_1(\tilde{T}_{\text{IMM}}(\mathcal{C})) \leq \text{cost}_1(\mathcal{C}) + \sum_{u \in T'_{\text{IMM}}(\mathcal{C})} t_u \cdot \|\mu'_u - \mu''_u\|_1.$$

The second step is the more involved part of the proof, and the part where the graph-based analysis presented here departs from [33].

**Lemma A.10** (Lemma 5.6 in [33])**.** *Let $H$ be the height of $\tilde{T}_{\text{IMM}}(\mathcal{C})$. Then,*

$$\sum_{u \in T'_{\text{IMM}}(\mathcal{C})} t_u \cdot \|\mu'_u - \mu''_u\|_1 \leq H \cdot \text{cost}_1(\tilde{T}_{\text{IMM}}(\mathcal{C})).$$

Since $\tilde{T}_{\text{IMM}}(\mathcal{C})$ has $k$ leaves, its height is at most $k$, thus Theorems A.9 and A.10 together immediately imply Theorem A.8.

---

[7]As mentioned in Section 1.1, for randomized algorithms, a tight bound of $(1 + o(1)) \log k$ is known.

*Proof of Theorem A.10.* Fix a node $u$ in $\tilde{T}_{\text{IMM}}(\mathcal{C})$. Let $\times_{j=1}^d [a_j^u, b_j^u]$ be a bounding box containing all of $Y^u$. Define the following distribution $\mathcal{D}$ over coordinate cuts $j, \tau$: first, pick $j \in \{1, \dots, d\}$ with probability proportional to $|b_j^u - a_j^u|$; then, pick $\tau$ uniformly at random in $[a_j^u, b_j^u]$.

Let $\tilde{G}(Y^u, \tilde{E})$ be any undirected graph over $Y^u$. The probability that an edge $xy \in \tilde{E}$ is cut by $(j, \tau) \sim \mathcal{D}$ is

$$\sum_{j=1}^d \frac{|b_j^u - a_j^u|}{\|b^u - a^u\|_1} \cdot \frac{|x_j - y_j|}{|b_j^u - a_j^u|} = \frac{\|x - y\|_1}{\|b^u - a^u\|_1},$$

and therefore,

$$\mathbb{E}_{(j,\tau) \sim \mathcal{D}} \left[ e_{\tilde{G}}(S_{j,\tau}, Y_u \setminus S_{j,\tau}) \right] = \frac{1}{\|b^u - a^u\|_1} \sum_{xy \in \tilde{E}} \|x - y\|_1. \tag{16}$$

Let $\mathcal{S}_u$ be the set of coordinate cuts $(j, \tau)$ that satisfy $\tau \in [a_j, b_j]$. Let $m_u(j, \tau)$ denote the number of mistakes that a cut $j, \tau$ makes (i.e., the number of points in $x \in X^u$ such that $c(x) \in M^u$, but the cut separates $x$ and $c(x)$). Recall that our modified IMM chooses the cut $j_u, \tau_u$ in the node among the cuts in $\mathcal{S}_u$ that separate the pair $\mu_u', \mu_u''$, and $t_u = m_u(j_u, \tau_u)$.

Let $G(Y^u, E_G)$ be the star graph on $Y^u$, where each $x \in X^u$ is adjacent to $c(x)$ if $c(x) \in M^u$, and is an isolated node otherwise. Observe that $m_u(j, \tau) = e_G(S_{j,\tau}, Y_u \setminus S_{j,\tau})$. Let $H(Y^u, E_H)$ be the graph that contains an single edge between $\mu_u', \mu_u''$. Observe that $e_H(S_{j,\tau}, Y_u \setminus S_{j,\tau})$ equals 1 if the cut $j, \tau$ separates $\mu_u', \mu_u''$, and 0 otherwise. We denote this as $e_H(S_{j,\tau}, Y_u \setminus S_{j,\tau}) = \mathbf{1}\{(j, \tau) \text{ separates } \mu_u', \mu_u''\}$. Therefore,

$$\begin{aligned}
t_u = m_u(j_u, \tau_u) &= \min_{\substack{j, \tau \in \mathcal{S}_u: \\ (j, \tau) \text{ separates } \mu_u', \mu_u''}} m_u(j, \tau) \\
&= \min_{j, \tau \in \mathcal{S}_u} \frac{m_u(j, \tau)}{\mathbf{1}\{(j, \tau) \text{ separates } \mu_u', \mu_u''\}} \\
&= \min_{j, \tau \in \mathcal{S}_u} \frac{e_G(S_{j,\tau}, Y_u \setminus S_{j,\tau})}{e_H(S_{j,\tau}, Y_u \setminus S_{j,\tau})} \\
&\leq \frac{\mathbb{E}_{(j,\tau) \sim \mathcal{D}} \left[ e_G(S_{j,\tau}, Y_u \setminus S_{j,\tau}) \right]}{\mathbb{E}_{(j,\tau) \sim \mathcal{D}} \left[ e_H(S_{j,\tau}, Y_u \setminus S_{j,\tau}) \right]} && \text{Theorem A.2} \\
&= \frac{\sum_{xy \in E_G} \|x - y\|_1}{\sum_{xy \in E_H} \|x - y\|_1} && \text{Equation (16)} \\
&= \frac{\sum_{x \in X^u} \|x - c(x)\|_1}{\|\mu_u' - \mu_u''\|_1} && \text{definition of } G \text{ and } H.
\end{aligned}$$

Rearranging, $t_u \cdot \|\mu_u' - \mu_u''\|_1 \leq \sum_{x \in X^u} \|x - c(x)\|_1$. Since in each level $L$ in the tree the clusters $\{X^u : u \in L\}$ form a partition of $X$, we get

$$\sum_{u \in L} t_u \cdot \|\mu_u' - \mu_u''\|_1 \leq \sum_{u \in L} \sum_{x \in X^u} \|x - c(x)\|_1 = \sum_{x \in X} \|x - c(x)\|_1 = \text{cost}_1(\mathcal{C}).$$

Summing again over the $H$ levels in the tree yields the lemma. $\qquad\square$

Note that the ratio $\frac{e_G(S_{j,\tau}, Y_u \setminus S_{j,\tau})}{e_H(S_{j,\tau}, Y_u \setminus S_{j,\tau})}$ that arises in the proof is the non-uniform cut sparsity $\Psi_{G,H}(S_{j,\tau})$ from Section 3.

## B Additional Experimental Results

**Running times.** Table 4 contains running time measurements. For reference-based methods (EMN, CART and SPEX-Clique) we measure the reference clustering step and tree construction step separately. For SPEX-kNN, we measure the $k$-NN graph construction step and the tree construction step separately.

In our experiments, Kernel IMM proved feasible to run only on the smaller datasets (R15, Pathbased, Iris, Ecoli, Breast Cancer). Each algorithm was allotted three hours to run on each dataset. On the

Table 4: Runtime comparison of the non kernel k-means based algorithms. Presented times are the median runtime across five runs. For the Clique and CART algorithms, performance was measured based on a spectral reference.

| Algorithm | Beans | Iris | CIFAR | Caltech 101 | R15 | Pathbased | Ecoli | Cancer | Newsgroups | MNIST |
|---|---|---|---|---|---|---|---|---|---|---|
| Spectral reference | 1.8s | 22.1ms | 1m12s | 7.6s | 91.6ms | 29ms | 34.2ms | 34.1ms | 8.54s | 1min53s |
| k-means reference | 86.4ms | 4.21ms | 2.69s | 2.1s | 10.4ms | 4.16ms | 5.39ms | 7.01ms | 1.04s | 5.16s |
| $k$-NN graph build | 1.87s | 7.01ms | 1m55s | 2.23s | 19ms | 3.57ms | 9.31ms | 8.03ms | 5.79s | 2min59s |
| SPEX-kNN | 3.69s | 27ms | 5m53s | 1m48s | 93.4ms | 22.8ms | 85.3ms | 90.8ms | 2min27s | 2min52s |
| SPEX-Clique | 203ms | 8.36ms | 28.2s | 1m12s | 17ms | 9.65ms | 13.3ms | 16.3ms | 13.2s | 39.4s |
| EMN | 13.5s | 6.51ms | 5m13s | 15m40s | 101ms | 13.3ms | 20.9ms | 184ms | 7min56s | 5min20s |
| CART | 349ms | 14.8ms | 32.2s | 8min51s | 72.7ms | 9.4ms | 33.9ms | 31.9ms | 39.2s | 30s |

Table 5: Performance of SPEX-kNN for different $k$ values. The reference compared in the REF column is the reference with the same $k$ applied.

| | Ecoli | | | Iris | | | Cancer | | |
|---|---|---|---|---|---|---|---|---|---|
| $k$ | ARI | AMI | REF | RS | AMI | REF | ARI | AMI | REF |
| 2 | .594 | .571 | .006 | .287 | .370 | .020 | .681 | .603 | -.019 |
| 5 | .594 | .589 | .640 | .450 | .647 | .895 | .594 | .546 | .688 |
| 10 | .682 | .648 | .828 | .450 | .647 | .514 | .507 | .490 | .547 |
| 15 | .682 | .648 | .854 | .450 | .647 | .445 | .507 | .490 | .567 |
| 20 | .679 | .642 | .863 | .450 | .647 | .450 | .507 | .490 | .562 |
| 50 | .679 | .638 | .593 | .600 | .642 | .756 | .507 | .490 | .532 |

larger datasets, Kernel IMM either ran out of memory, or failed to complete running within the allotted time. All other methods finished running within up to 16 minutes (see Table 4). Therefore, we report results for Kernel IMM only for the smaller datasets.

**kNN graph parameters.** Table 5 includes additional results for SPEX-kNN, showing how its performance changes as $k$ (the parameter of the $k$-NN graph) varies on the Ecoli, Iris and Breast Cancer datasets.

**Number of leaves.** As mentioned in Section 2, SPEX can produce a tree with any desired number of leaves $\ell$. Our evaluation so far has focused on the setting $\ell = k$, i.e., the number of leaves in the output tree is equal to the number of clusters in the reference clustering. To clarify the relation between $k$ and $\ell$, viewing the reference clustering as a trained model, $k$ is its number of outcomes and $\ell$ is the number of explanations our explainability method can yield. Since each outcome must have its own separate explanation, we must have $\ell \geq k$, thus $\ell = k$ is the "most explainable" setting. As $\ell$ increases, the expressiveness of the explainable clustering tree grows and it is better able to approximate the reference clustering, albeit at the cost of explainability, since now some model outcomes would have multiple different explanations.

Table 6 includes result with $\ell = 2k$ for SPEX-Clique, CART-Spectral, CART-$k$-means, and ExKMC [17], which is an extension of IMM to support more leaves (which IMM does not naturally support, due to its reliance on reference clustering centroids; the same goes for EMN). The results do not point to a clearly superior method.

## B.1 Weighted CART

The implementation of CART in Scikit-Learn [36] contains a weighted variant different from the standard one described in Section 3.3. It has a modified selection rule for the next leaf to split. When the stopping condition is a pre-specified number of leaves it is the default mode. It performs significantly better than usual CART on some datasets. As we find this noteworthy, the results are reported here. A similar modification can be applied to SPEX-Clique, and it is reported too. The results are in Tables 7 and 8.

Table 6: Results with $\ell = 2k$ leaves.

| Algorithm | Ecoli ARI | Ecoli AMI | Iris ARI | Iris AMI | Cancer ARI | Cancer AMI | MNIST ARI | MNIST AMI | Caltech 101 ARI | Caltech 101 AMI | Newsgroups ARI | Newsgroups AMI | Beans ARI | Beans AMI | Cifar ARI | Cifar AMI |
|---|---|---|---|---|---|---|---|---|---|---|---|---|---|---|---|---|
| SpEx-Clique | .469 | .589 | .716 | .739 | .491 | .464 | .197 | .317 | .203 | .565 | .088 | .281 | .370 | .512 | .309 | .484 |
| ExKMC | .458 | .582 | .730 | .755 | .491 | .464 | .224 | .328 | .249 | .546 | .096 | .263 | .370 | .512 | .323 | .467 |
| CART-$k$-means | .447 | .573 | .716 | .739 | .491 | .464 | .031 | .177 | 0 | .208 | .026 | .148 | .370 | .512 | .144 | .371 |
| CART-Spectral | .434 | .493 | .610 | .648 | .749 | .631 | .080 | .210 | 0 | .214 | .006 | .104 | .608 | .695 | 135 | .381 |

Table 7: Results on smaller datasets.

| Algorithm | R15 ARI | R15 AMI | R15 REF | Pathbased ARI | Pathbased AMI | Pathbased REF | Ecoli ARI | Ecoli AMI | Ecoli REF | Iris ARI | Iris AMI | Iris REF | Cancer ARI | Cancer AMI | Cancer REF |
|---|---|---|---|---|---|---|---|---|---|---|---|---|---|---|---|
| *REF:* Spectral | .993 | .994 | 1. | .526 | .570 | 1. | .711 | .653 | 1. | .630 | .661 | 1. | .743 | .626 | 1. |
| SpEx-kNN | .982 | .987 | .989 | .332 | .410 | .551 | **.679** | **.642** | .863 | .450 | **.647** | .450 | .507 | .490 | .562 |
| SpEx-Clique | **.986** | **.989** | **.993** | **.441** | **.517** | **.824** | .662 | .621 | .847 | **.576** | .629 | **.787** | **.694** | **.588** | .785 |
| SpEx-Clique-modified | **.986** | **.989** | **.993** | **.441** | **.517** | **.824** | .672 | .618 | **.886** | **.576** | .629 | **.787** | .683 | .560 | **.811** |
| CART | **.986** | **.989** | **.993** | **.441** | **.517** | **.824** | .672 | .618 | **.886** | **.576** | .629 | **.787** | .683 | .560 | **.811** |
| CART-modified | **.986** | **.989** | **.993** | **.441** | **.517** | **.824** | .672 | .618 | **.886** | **.576** | .629 | **.787** | .683 | .560 | **.811** |
| *REF:* $k$-means | .993 | .994 | 1. | .461 | .543 | 1. | .489 | .609 | 1. | .641 | .669 | 1. | .491 | .464 | 1. |
| EMN | **.986** | **.989** | **.993** | **.461** | **.543** | **1.** | **.456** | **.559** | **.873** | **.576** | **.629** | **.772** | **.491** | **.464** | **1.** |
| SpEx-Clique | **.986** | **.989** | **.993** | **.461** | **.543** | **1.** | **.456** | **.559** | **.873** | **.576** | **.629** | **.772** | **.491** | **.464** | **1.** |
| SpEx-Clique-modified | **.986** | **.989** | **.993** | **.461** | **.543** | **1.** | **.456** | **.559** | **.873** | **.576** | **.629** | **.772** | **.491** | **.464** | **1.** |
| CART | **.986** | **.989** | **.993** | .421 | .507 | .897 | .454 | .543 | .840 | **.576** | **.629** | **.772** | **.491** | **.464** | **1.** |
| CART-modified | **.986** | **.989** | **.993** | **.461** | **.543** | **1.** | **.456** | **.559** | **.873** | **.576** | **.629** | **.772** | **.491** | **.464** | **1.** |
| *REF:* Kernel $k$-means | .908 | .967 | 1. | .919 | .888 | 1. | .538 | .612 | 1. | .731 | .767 | 1. | .119 | .230 | 1. |
| Kernel IMM | **.904** | **.962** | **.986** | **.614** | **.614** | **.583** | .522 | .560 | .848 | **.732** | **.788** | **.924** | .127 | .241 | **.930** |
| SpEx-Clique | .869 | .941 | .951 | .479 | .553 | .450 | **.529** | **.573** | **.851** | **.732** | **.788** | **.924** | **.406** | **.414** | .516 |
| SpEx-Clique-modified | .872 | .944 | .954 | .479 | .553 | .450 | **.529** | **.573** | **.851** | **.732** | **.788** | **.924** | **.406** | **.414** | .516 |
| CART | .682 | .876 | .759 | .479 | .553 | .450 | .500 | .558 | .824 | **.732** | **.788** | **.924** | **.406** | **.414** | .516 |
| CART-modified | .880 | **.944** | **.955** | .479 | .553 | .450 | .500 | .558 | .824 | **.732** | **.788** | **.924** | **.406** | **.414** | .516 |


Table 8: Results on larger datasets.

| Algorithm | MNIST | | | Caltech 101 | | | Newsgroups | | | Beans | | | Cifar | | |
|---|---|---|---|---|---|---|---|---|---|---|---|---|---|---|---|
| | ARI | AMI | REF | ARI | AMI | REF | ARI | AMI | REF | ARI | AMI | REF | ARI | AMI | REF |
| ***REF:*** Spectral | .745 | .820 | 1. | .563 | .859 | 1. | .431 | .671 | 1. | .586 | .677 | 1. | .712 | .801 | 1. |
| SPEx-kNN | .092 | .338 | .150 | .121 | .497 | .168 | .042 | .170 | .090 | **.574** | **.690** | **.649** | .342 | .434 | .373 |
| SPEx-Clique | .217 | **.384** | .282 | .247 | **.521** | **.303** | .078 | .223 | .189 | .564 | .671 | **.743** | .320 | **.438** | **.394** |
| SpEx-Clique-modified | .018 | .155 | .040 | .006 | .330 | .032 | .021 | .173 | .021 | .152 | .374 | .200 | .152 | .374 | .200 |
| CART | .027 | .148 | .030 | -.010 | .166 | .005 | .008 | .078 | -.013 | .036 | .169 | .035 | .036 | .169 | .035 |
| CART-modified | **.279** | .379 | **.321** | **.219** | .514 | .302 | **.106** | **.244** | **.223** | .380 | .448 | .418 | **.380** | **.448** | **.418** |
| ***REF:*** $k$-means | .364 | .481 | 1. | .405 | .822 | 1. | .502 | .660 | 1. | .572 | .689 | 1. | .636 | .738 | 1. |
| EMN | **.250** | **.342** | **.420** | **.249** | **.548** | **.416** | **.115** | .219 | **.163** | **.563** | **.688** | **.780** | .314 | .402 | **.387** |
| SPEx-Clique | .209 | .336 | .403 | .122 | .495 | .195 | .098 | **.249** | .128 | .562 | .687 | .773 | .288 | **.410** | .370 |
| SpEx-Clique-modified | .151 | .339 | .284 | .039 | .373 | .053 | .005 | .092 | .006 | .170 | .341 | .200 | .170 | .341 | .200 |
| CART | .124 | .299 | .229 | -.017 | .082 | .003 | .005 | .092 | .006 | .045 | .180 | .037 | .045 | .180 | .037 |
| CART-modified | .232 | .315 | .418 | .136 | .507 | .219 | .106 | .244 | .128 | .330 | .433 | .372 | **.330** | **.433** | .372 |

- The answer NA means that the paper has no limitation while the answer No means that the paper has limitations, but those are not discussed in the paper.
- The authors are encouraged to create a separate "Limitations" section in their paper.
- The paper should point out any strong assumptions and how robust the results are to violations of these assumptions (e.g., independence assumptions, noiseless settings, model well-specification, asymptotic approximations only holding locally). The authors should reflect on how these assumptions might be violated in practice and what the implications would be.
- The authors should reflect on the scope of the claims made, e.g., if the approach was only tested on a few datasets or with a few runs. In general, empirical results often depend on implicit assumptions, which should be articulated.
- The authors should reflect on the factors that influence the performance of the approach. For example, a facial recognition algorithm may perform poorly when image resolution is low or images are taken in low lighting. Or a speech-to-text system might not be used reliably to provide closed captions for online lectures because it fails to handle technical jargon.
- The authors should discuss the computational efficiency of the proposed algorithms and how they scale with dataset size.
- If applicable, the authors should discuss possible limitations of their approach to address problems of privacy and fairness.
- While the authors might fear that complete honesty about limitations might be used by reviewers as grounds for rejection, a worse outcome might be that reviewers discover limitations that aren't acknowledged in the paper. The authors should use their best judgment and recognize that individual actions in favor of transparency play an important role in developing norms that preserve the integrity of the community. Reviewers will be specifically instructed to not penalize honesty concerning limitations.

3. **Theory assumptions and proofs**

   Question: For each theoretical result, does the paper provide the full set of assumptions and a complete (and correct) proof?

   Answer: [Yes]

   Justification: Full rigorous proofs are provided in the appendix

   Guidelines:

   - The answer NA means that the paper does not include theoretical results.
   - All the theorems, formulas, and proofs in the paper should be numbered and cross-referenced.
   - All assumptions should be clearly stated or referenced in the statement of any theorems.
   - The proofs can either appear in the main paper or the supplemental material, but if they appear in the supplemental material, the authors are encouraged to provide a short proof sketch to provide intuition.

- Inversely, any informal proof provided in the core of the paper should be complemented by formal proofs provided in appendix or supplemental material.
- Theorems and Lemmas that the proof relies upon should be properly referenced.

4. **Experimental result reproducibility**

Question: Does the paper fully disclose all the information needed to reproduce the main experimental results of the paper to the extent that it affects the main claims and/or conclusions of the paper (regardless of whether the code and data are provided or not)?

Answer: [Yes]

Justification: All information needed for reproducibility is included in the paper; experimental code is also enclosed in supplementary material

Guidelines:

- The answer NA means that the paper does not include experiments.
- If the paper includes experiments, a No answer to this question will not be perceived well by the reviewers: Making the paper reproducible is important, regardless of whether the code and data are provided or not.
- If the contribution is a dataset and/or model, the authors should describe the steps taken to make their results reproducible or verifiable.
- Depending on the contribution, reproducibility can be accomplished in various ways. For example, if the contribution is a novel architecture, describing the architecture fully might suffice, or if the contribution is a specific model and empirical evaluation, it may be necessary to either make it possible for others to replicate the model with the same dataset, or provide access to the model. In general. releasing code and data is often one good way to accomplish this, but reproducibility can also be provided via detailed instructions for how to replicate the results, access to a hosted model (e.g., in the case of a large language model), releasing of a model checkpoint, or other means that are appropriate to the research performed.
- While NeurIPS does not require releasing code, the conference does require all submissions to provide some reasonable avenue for reproducibility, which may depend on the nature of the contribution. For example
  (a) If the contribution is primarily a new algorithm, the paper should make it clear how to reproduce that algorithm.
  (b) If the contribution is primarily a new model architecture, the paper should describe the architecture clearly and fully.
  (c) If the contribution is a new model (e.g., a large language model), then there should either be a way to access this model for reproducing the results or a way to reproduce the model (e.g., with an open-source dataset or instructions for how to construct the dataset).
  (d) We recognize that reproducibility may be tricky in some cases, in which case authors are welcome to describe the particular way they provide for reproducibility. In the case of closed-source models, it may be that access to the model is limited in some way (e.g., to registered users), but it should be possible for other researchers to have some path to reproducing or verifying the results.

5. **Open access to data and code**

Question: Does the paper provide open access to the data and code, with sufficient instructions to faithfully reproduce the main experimental results, as described in supplemental material?

Answer: [Yes]

Justification: All datasets used are publicly available and clearly referenced. Our code is included in the supplementary and will be made available online with the paper.

Guidelines:

- The answer NA means that paper does not include experiments requiring code.
- Please see the NeurIPS code and data submission guidelines (https://nips.cc/public/guides/CodeSubmissionPolicy) for more details.

- While we encourage the release of code and data, we understand that this might not be possible, so "No" is an acceptable answer. Papers cannot be rejected simply for not including code, unless this is central to the contribution (e.g., for a new open-source benchmark).
- The instructions should contain the exact command and environment needed to run to reproduce the results. See the NeurIPS code and data submission guidelines (https://nips.cc/public/guides/CodeSubmissionPolicy) for more details.
- The authors should provide instructions on data access and preparation, including how to access the raw data, preprocessed data, intermediate data, and generated data, etc.
- The authors should provide scripts to reproduce all experimental results for the new proposed method and baselines. If only a subset of experiments are reproducible, they should state which ones are omitted from the script and why.
- At submission time, to preserve anonymity, the authors should release anonymized versions (if applicable).
- Providing as much information as possible in supplemental material (appended to the paper) is recommended, but including URLs to data and code is permitted.

6. **Experimental setting/details**

Question: Does the paper specify all the training and test details (e.g., data splits, hyper-parameters, how they were chosen, type of optimizer, etc.) necessary to understand the results?

Answer: [Yes]

Justification: All parameter settings are described in the text.

Guidelines:

- The answer NA means that the paper does not include experiments.
- The experimental setting should be presented in the core of the paper to a level of detail that is necessary to appreciate the results and make sense of them.
- The full details can be provided either with the code, in appendix, or as supplemental material.

7. **Experiment statistical significance**

Question: Does the paper report error bars suitably and correctly defined or other appropriate information about the statistical significance of the experiments?

Answer: [NA]

Justification: All methods evaluated in this paper (both our methods and the baseline) are deterministic algorithms.

Guidelines:

- The answer NA means that the paper does not include experiments.
- The authors should answer "Yes" if the results are accompanied by error bars, confidence intervals, or statistical significance tests, at least for the experiments that support the main claims of the paper.
- The factors of variability that the error bars are capturing should be clearly stated (for example, train/test split, initialization, random drawing of some parameter, or overall run with given experimental conditions).
- The method for calculating the error bars should be explained (closed form formula, call to a library function, bootstrap, etc.)
- The assumptions made should be given (e.g., Normally distributed errors).
- It should be clear whether the error bar is the standard deviation or the standard error of the mean.
- It is OK to report 1-sigma error bars, but one should state it. The authors should preferably report a 2-sigma error bar than state that they have a 96% CI, if the hypothesis of Normality of errors is not verified.
- For asymmetric distributions, the authors should be careful not to show in tables or figures symmetric error bars that would yield results that are out of range (e.g. negative error rates).

- If error bars are reported in tables or plots, The authors should explain in the text how they were calculated and reference the corresponding figures or tables in the text.

8. **Experiments compute resources**

   Question: For each experiment, does the paper provide sufficient information on the computer resources (type of compute workers, memory, time of execution) needed to reproduce the experiments?

   Answer: [Yes]

   Justification: Our experiments are run on a standard regular free-tier cloud CPU machine (Google Colab) and can be reproduced on any similar machine; our methods and experiments do not require special computational resources. No additional compute beyond what is described in the paper was used in the course of preparing this paper.

   Guidelines:
   - The answer NA means that the paper does not include experiments.
   - The paper should indicate the type of compute workers CPU or GPU, internal cluster, or cloud provider, including relevant memory and storage.
   - The paper should provide the amount of compute required for each of the individual experimental runs as well as estimate the total compute.
   - The paper should disclose whether the full research project required more compute than the experiments reported in the paper (e.g., preliminary or failed experiments that didn't make it into the paper).

9. **Code of ethics**

   Question: Does the research conducted in the paper conform, in every respect, with the NeurIPS Code of Ethics https://neurips.cc/public/EthicsGuidelines?

   Answer: [Yes]

   Justification: Code of Ethics was fully observed.

   Guidelines:
   - The answer NA means that the authors have not reviewed the NeurIPS Code of Ethics.
   - If the authors answer No, they should explain the special circumstances that require a deviation from the Code of Ethics.
   - The authors should make sure to preserve anonymity (e.g., if there is a special consideration due to laws or regulations in their jurisdiction).

10. **Broader impacts**

    Question: Does the paper discuss both potential positive societal impacts and negative societal impacts of the work performed?

    Answer: [Yes]

    Justification: An impact statement is included as part of the manuscript with an appropriate discussion of potential broader impact.

    Guidelines:
    - The answer NA means that there is no societal impact of the work performed.
    - If the authors answer NA or No, they should explain why their work has no societal impact or why the paper does not address societal impact.
    - Examples of negative societal impacts include potential malicious or unintended uses (e.g., disinformation, generating fake profiles, surveillance), fairness considerations (e.g., deployment of technologies that could make decisions that unfairly impact specific groups), privacy considerations, and security considerations.
    - The conference expects that many papers will be foundational research and not tied to particular applications, let alone deployments. However, if there is a direct path to any negative applications, the authors should point it out. For example, it is legitimate to point out that an improvement in the quality of generative models could be used to generate deepfakes for disinformation. On the other hand, it is not needed to point out that a generic algorithm for optimizing neural networks could enable people to train models that generate Deepfakes faster.

- The authors should consider possible harms that could arise when the technology is being used as intended and functioning correctly, harms that could arise when the technology is being used as intended but gives incorrect results, and harms following from (intentional or unintentional) misuse of the technology.
- If there are negative societal impacts, the authors could also discuss possible mitigation strategies (e.g., gated release of models, providing defenses in addition to attacks, mechanisms for monitoring misuse, mechanisms to monitor how a system learns from feedback over time, improving the efficiency and accessibility of ML).

11. **Safeguards**

Question: Does the paper describe safeguards that have been put in place for responsible release of data or models that have a high risk for misuse (e.g., pretrained language models, image generators, or scraped datasets)?

Answer: [NA]

Justification: No new data or models are released with the paper; the paper poses no such risks.

Guidelines:

- The answer NA means that the paper poses no such risks.
- Released models that have a high risk for misuse or dual-use should be released with necessary safeguards to allow for controlled use of the model, for example by requiring that users adhere to usage guidelines or restrictions to access the model or implementing safety filters.
- Datasets that have been scraped from the Internet could pose safety risks. The authors should describe how they avoided releasing unsafe images.
- We recognize that providing effective safeguards is challenging, and many papers do not require this, but we encourage authors to take this into account and make a best faith effort.

12. **Licenses for existing assets**

Question: Are the creators or original owners of assets (e.g., code, data, models), used in the paper, properly credited and are the license and terms of use explicitly mentioned and properly respected?

Answer: [Yes]

Justification: All uses of existing assets (datasets, as well as images reproduced based on prior works) are fully and properly credited.

Guidelines:

- The answer NA means that the paper does not use existing assets.
- The authors should cite the original paper that produced the code package or dataset.
- The authors should state which version of the asset is used and, if possible, include a URL.
- The name of the license (e.g., CC-BY 4.0) should be included for each asset.
- For scraped data from a particular source (e.g., website), the copyright and terms of service of that source should be provided.
- If assets are released, the license, copyright information, and terms of use in the package should be provided. For popular datasets, paperswithcode.com/datasets has curated licenses for some datasets. Their licensing guide can help determine the license of a dataset.
- For existing datasets that are re-packaged, both the original license and the license of the derived asset (if it has changed) should be provided.
- If this information is not available online, the authors are encouraged to reach out to the asset's creators.

13. **New assets**

Question: Are new assets introduced in the paper well documented and is the documentation provided alongside the assets?

Answer: [NA]

Justification: No new assets released.

Guidelines:

- The answer NA means that the paper does not release new assets.
- Researchers should communicate the details of the dataset/code/model as part of their submissions via structured templates. This includes details about training, license, limitations, etc.
- The paper should discuss whether and how consent was obtained from people whose asset is used.
- At submission time, remember to anonymize your assets (if applicable). You can either create an anonymized URL or include an anonymized zip file.

14. **Crowdsourcing and research with human subjects**

Question: For crowdsourcing experiments and research with human subjects, does the paper include the full text of instructions given to participants and screenshots, if applicable, as well as details about compensation (if any)?

Answer: [NA]

Justification: The paper does not involve crowdsourcing nor research with human subjects.

Guidelines:

- The answer NA means that the paper does not involve crowdsourcing nor research with human subjects.
- Including this information in the supplemental material is fine, but if the main contribution of the paper involves human subjects, then as much detail as possible should be included in the main paper.
- According to the NeurIPS Code of Ethics, workers involved in data collection, curation, or other labor should be paid at least the minimum wage in the country of the data collector.

15. **Institutional review board (IRB) approvals or equivalent for research with human subjects**

Question: Does the paper describe potential risks incurred by study participants, whether such risks were disclosed to the subjects, and whether Institutional Review Board (IRB) approvals (or an equivalent approval/review based on the requirements of your country or institution) were obtained?

Answer: [NA]

Justification: paper does not involve crowdsourcing nor research with human subjects.

Guidelines:

- The answer NA means that the paper does not involve crowdsourcing nor research with human subjects.
- Depending on the country in which research is conducted, IRB approval (or equivalent) may be required for any human subjects research. If you obtained IRB approval, you should clearly state this in the paper.
- We recognize that the procedures for this may vary significantly between institutions and locations, and we expect authors to adhere to the NeurIPS Code of Ethics and the guidelines for their institution.
- For initial submissions, do not include any information that would break anonymity (if applicable), such as the institution conducting the review.

16. **Declaration of LLM usage**

Question: Does the paper describe the usage of LLMs if it is an important, original, or non-standard component of the core methods in this research? Note that if the LLM is used only for writing, editing, or formatting purposes and does not impact the core methodology, scientific rigorousness, or originality of the research, declaration is not required.

Answer: [NA]

Justification: the core method development in this research does not involve LLMs as any important, original, or non-standard components.

Guidelines:

- The answer NA means that the core method development in this research does not involve LLMs as any important, original, or non-standard components.
- Please refer to our LLM policy (https://neurips.cc/Conferences/2025/LLM) for what should or should not be described.

