# OpenReview forum: "SpEx: A Spectral Approach to Explainable Clustering"
_NeurIPS.cc/2025/Conference — NeurIPS 2025 poster_

### Official Review · Reviewer_cfLS · 2025-07-01

**Clarity:** 3
**Significance:** 3
**Originality:** 3
**Rating:** 4
**Confidence:** 4

**Summary:**

The paper presents SpEx, an explainable approach for spectral clustering. Specifically, the paper considers a generic approach that is based on spectral graph clustering paired with an iterative tree construction algorithm. The paper investigates two variants of the approach: SpEx-Clique that can represent any reference clustering using a clique graph and SpEx-kNN that does not require a reference clustering and is based on the nearest neighbours in the dataset. Experiments on eight datasets show that SpEx-Clique outperforms the interpretable baselines.

**Questions:**

Please see my main concerns above. No further questions.

**Ethical Concerns:**

["NO or VERY MINOR ethics concerns only"]

**Final Justification:**

As I noted in my review there are highly-related works are missing from both literature review and as baselines (including one that seems directly comparable). I think the paper would significantly benefit from extended literature review covering related works and comparison with relevant baselines. I will therefore maintain my borderline accept score.

**Limitations:**

yes

**Paper Formatting Concerns:**

No issues

**Quality:**

2

**Strengths And Weaknesses:**

Strengths:
- Interesting topic that has received significant attention recently
- Strong theoretical basis including Theorem 2.2 and the generalized framework based on non-uniform sparse cuts
- Experiments show the proposed approach (specifically, SpEx-Clique) outperforms the baselines

----

Weaknesses:
- The paper claims that the proposed approach "has the flexibility to produce a tree T with any desired number of leaves, regardless of the number of clusters in the reference clustering” in contrast to approaches like IMM and EMN. It is not entirely clear to me why this is a benefit, as the other approaches can simply increase the number of clusters in the algorithm that produces the reference clustering. In contrast, my understanding of the proposed approach is that it still suffers from the main limitation of IMM and EMN where one leaf corresponds to one cluster hence, adding additional leaves forces the use of additional clusters in the resulting partition.

- Highly-related works are missing from both literature review and as baselines:
	* There are existing works highly-related to spectral clustering that are not mentioned or considered as baselines. Specifically, [1] is an optimization-based approach that can support various clustering objectives and includes experiments with both kernel and spectral objectives, [2] which is cited, includes a generic approach that can be used with non-Euclidean objectives and was used Spherical K-Means objective in [2] and with spectral objective in [3]; [3] focused on interpretable clustering with alternative interpretable model (generalized additive model) and has been tested with SpectralNet embeddings; [4] is an interpretable Kernel K-Means approach (based on decision trees) that does not require centroids.
	* The widely used ExKMC algorithm [5] (that can utilize multiple leaves per cluster) is cited but is not used as a baseline.

- Experimental results:
	* Experiments only compare the approach to the baselines in terms of external evaluation (ARI and AMI are based on labels), however it would be very interesting to report both “the cost of interpretability” in the proposed approach and also the suboptimality as a result of using a greedy optimization procedure.
	* It would be very useful to report aggregate metrics across datasets like average ARI and AMI to better understand the performance across datasets, ideally with standard deviation.
	* While results show that SpEx-Clique outperforms the baselines, the gains seem somewhat limited and varies across datasets.

- Minor:
- Line 141: “graph nodes graph”
- what distance metric is used for the kNN graph?

----

[1] Cohen, E. (2023, May). Interpretable clustering via soft clustering trees. In International Conference on Integration of Constraint Programming, Artificial Intelligence, and Operations Research (pp. 281-298). Cham: Springer Nature Switzerland.

[2] Gabidolla, M., & Carreira-Perpiñán, M. Á. (2022, August). Optimal interpretable clustering using oblique decision trees. In Proceedings of the 28th ACM SIGKDD Conference on Knowledge Discovery and Data Mining (pp. 400-410).

[3] Upadhya, N., & Cohen, E. (2024, January). NeurCAM: Interpretable Neural Clustering via Additive Models. In ECAI.

[4] Ohl, L., Mattei, P. A., Leclercq, M., Droit, A., & Precioso, F. (2024). Kernel KMeans clustering splits for end-to-end unsupervised decision trees. arXiv preprint arXiv:2402.12232.

[5] Frost, N., Moshkovitz, M., & Rashtchian, C. (2020). ExKMC: Expanding Explainable $ k $-Means Clustering. arXiv preprint arXiv:2006.02399.

---

> ### Author Rebuttal · Authors · 2025-07-29
>
> Thank you for your valuable review and support!
>
> > The paper claims that the proposed approach "has the flexibility to produce a tree T with any desired number of leaves, regardless of the number of clusters in the reference clustering” in contrast to approaches like IMM and EMN. It is not entirely clear to me why this is a benefit, as the other approaches can simply increase the number of clusters in the algorithm that produces the reference clustering.
>
> We did not mean that this flexibility is necessarily beneficial (we will change the wording to correct this impression). Increasing the number of leaves to more than $k$ means modifying the notion of explanability to a weaker one since the same class would have multiple different explanations. Thus it prioritizes accuracy over explainability which is a use-case dependent decision.
>
> What we meant to point out is that in SpEx this weaker explainability is possible by just partitioning more nodes with the same procedure. This is not possible with EMN for example, since its node partitioning procedure requires at least two reference centroids in the node, and is otherwise undefined (due to its score being divided by zero, see line 275).
>
> > Highly-related works are missing from both literature review and as baselines
>
> Thank you for the highly relevant references, we will include an appropriate discussion. We note that [1,2,3] use different and more relaxed notions of explainability and are not directly comparable. [4] seems compatible with a kernel $k$-means reference; we have not been able to find an implementation of this method but we will look into this further. [5] allows more leaves than our methods, $\ell>k$; we include a comparison in the response to K9ow.
>
> > it would be very interesting to report both “the cost of interpretability” in the proposed approach
>
> This can be read in the REF column of Tables 2 and 3 (at least under one definition of cost of explainability – abbrev. CoE – as there are a few ways to define this for general reference clusterings). The REF column treats the reference clustering as the ground truth clustering, entirely ignoring the given labels of the datasets (as opposed the ARI and AMI columns as well as Figure 3, which all measure clustering quality with respect to the given labels). In each band of rows (spectral, $k$-means and kernel $k$-means), the REF column of the top row (which contains the reference clustering) equals 1 definitionally. The other rows list the ARI of each explainable clustering method with respect to the reference clustering as ground truth. This is the inverse CoE (higher is better).
>
> The REF results show that for the spectral reference, SpEx-Clique has the best CoE (highest REF) overall (on all datasets except Ecoli and Cancer where it is slightly outperformed by CART). For the $k$-means reference, EMN has the best CoE – as expected, since EMN is tailored to the $k$-means reference and is provably near-optimal for it by [11]. Similarly, Kernel IMM often has the best CoE for kernel $k$-means, since it is tailored to it.
>
> > and also the suboptimality as a result of using a greedy optimization procedure
>
> It is not clear how to measure this since there is no clear way to compute the best explainable clustering tree overall. We mention, however, that prior work [11,18] has proved that greedy algorithms are worst-case near-optimal for explainable $k$-means.
>
> > It would be very useful to report aggregate metrics across datasets like average ARI and AMI to better understand the performance across datasets, ideally with standard deviation
>
> Thank you for the suggestion – we include aggregate metrics below. We report the median error across datasets with the [p25 ,p75] quantiles as confidence intervals rather than mean/std since our experimental setting is deterministic. The error is measured as the difference from the best reference clustering for each dataset to offset the different intrinsic clusterability across datasets. Lower is better.
>
> | | SpEx-Clique | SpEx-kNN | EMN | CART-k-means | CART-spectral |
> |---|---|---|---|---|---|
> | ARI error | $\mathbf{0.190}_{[\mathbf{.049},.4]}$ | $0.303_{[.151,.446]}$ | $0.284_{[.205,\mathbf{.390}]}$ | $0.377_{[.205,.577]}$ | $0.279_{[.056,.591]}$ |
> | AMI error | $\mathbf{0.187}_{[\mathbf{.036},\mathbf{.381}]}$ | $0.247_{[.192,.396]}$ | $0.235_{[.080,.412]}$ | $0.341_{[.092,.589]}$ | $0.330_{[.039,.642]}$ |

---

> > ### Comment · Reviewer_cfLS · 2025-08-04
> >
> > Thank you for your response and clarifications. I think the paper would significantly benefit from extended literature review covering related works and comparison with relevant baselines, but remain positive about the paper.

---

### Official Review · Reviewer_K9ow · 2025-07-01

**Clarity:** 3
**Significance:** 2
**Originality:** 3
**Rating:** 4
**Confidence:** 4

**Summary:**

This paper suggests a new approach to explainable clustering with decision trees, based on spectral graph theory. Concretely, the authors propose SPEX, an algorithm that constructs a decision tree based on a graph $G(X,E,w)$. The tree is constructed by iteratively choosing a leaf and an axis-aligned cut that maximize the decrease in conductance at that leaf. The goal is to greedily find a tree $T$ that minimizes a quantity $\bar{\Psi}_G(T)$ that can be viewed as a normalized cut objective for multi-way partitions. The paper justifies this choice of objective by adapting a theorem that, broadly speaking, asserts that axis-aligned cuts with low conductance exist if points that are connected in the graph through an edge are on average closer than random points. The SPEX algorithm can be used for any graph, in particular for the clique graph in which points from the same (reference) cluster are connected. Alternatively, one may also directly use the kNN graph as input. Building on this graph characterization of explainable clustering, the paper shows that previous algorithms based on reference clusters (slightly adapted variant of IMM, EMN, CART) can be viewed as minimizing different non-uniform sparse cuts. The paper also includes a set of experiments that measure Adjusted Rand Index (ARI) and Adjusted Mutual Information (AMI) between different explainable clustering methods and ground truth clusters (corresponding to labels).

**Questions:**

- It seems in Appendix Lemma A.10 you are actually also bounding the number of errors of a random cut. Can your analysis be used to say something about the price of Random Cut?

- What is the price of explainability for SPEX on k-means experiments? (I am mainly asking this because I feel it should be included, given that the methods you compare against aim to minimize that quantity).

- Did the authors consider comparing against other practical explainable clustering algorithms [1,2]?

**Ethical Concerns:**

["NO or VERY MINOR ethics concerns only"]

**Final Justification:**

After the author's response, I remain positive about the paper. To summarize, I feel this paper provides a valuable contribution to the line of work on explainable clustering. Further contextualizing the paper by adding some of the experiments from their rebuttal would strengthen the paper.

**Limitations:**

Yes

**Quality:**

3

**Strengths And Weaknesses:**

**Strengths**

This paper is a useful contribution to the literature on explainable clustering via decision trees. Most recent works restrict themselves to studying theoretical approximation guarantees for decision trees in k-means and k-medians (with an extension to kernel k-means with certain kernels). Posing the question in a graph framework is a new perspective. It generalizes some of the previous works, allows insights into what distinguishes some algorithms from others (Section 3 is interesting), and gives a solid motivation for the proposed algorithm. The method can take any graph as input, including kNN which in some applications is more meaningful than e.g. reference cluster centers obtained via k-means. Overall, the paper is written well.

**Weaknesses**

While the method itself is theoretically motivated (Theorem 2.2.), the authors do not use this result to give guarantees for their explainable clustering algorithm (beyond a small k-means derivation in Appendix A.2). It should also be noted that there are several papers on clustering with decision trees beyond theoretical studies on price of explainability, some of which also do not need reference centers, and CART is just one of them. For example, see [1,2 and references therein].

Minor comments and typos:
- The first figure seems to label the points according to the Gaussian they came from. Would it not be more reasonable in clustering to colour them according to some k-means partition?
- $e_G$ and $vol_G$ not defined.
- Line 249: graph-theortic

[1] Fraiman, Ricardo, Badih Ghattas, and Marcela Svarc. "Interpretable clustering using unsupervised binary trees." Advances in Data Analysis and Classification 7 (2013): 125-145.

[2] Frost, Nave, Michal Moshkovitz, and Cyrus Rashtchian. "ExKMC: Expanding Explainable $ k $-Means Clustering." arXiv preprint arXiv:2006.02399 (2020).

---

> ### Author Rebuttal · Authors · 2025-07-29
>
> Thank you for your valuable review and support!
>
> > It seems in Appendix Lemma A.10 you are actually also bounding the number of errors of a random cut. Can your analysis be used to say something about the price of Random Cut?
>
> Yes, we illustrate this below; however, we preface it by noting that the Random Cut algorithm has been extensively studied in theory [11, 17, 18, 28. 30] and tight bounds for its price of explainability are already known, even up to the leading constant. Hence our techniques do not lead to new results for this algorithm, and we do not claim a contribution in this vein.
>
> To illustrate how our analysis can be applied to Random Cut, we show it implies an $O(k)$ price-of-explainability bound for Random Cut, the same bound stated for IMM in Theorems A.7 and A.8. Consider Random Cut over the unweighted star graph for explainable $k$-medians. In a fixed node $u$, by a triangle inequality, a “mistake” (i.e., a point separated from its centroid) contributes at most an additive $\|b_u-a_u\|_1$ to the explainable clustering cost. In eq. (16) in Lemma A.10, the left-hand side is the expected number of mistakes of Random Cut and the right hand side is the reference clustering cost of centroids in $u$ divided by $\|b_u-a_u\|_1$. Hence, the expected additive loss in $u$ is at most the reference clustering cost of centroids in $u$. In each tree level the nodes induce a partition of the centroids, hence the expected additive loss of Random Cut per level is $O(cost_1(\mathcal C))$. Finally, summing over at most $k-1$ levels in the tree, the total additive loss in clustering cost is $O(k\cdot cost_1(\mathcal C))$.
>
> > What is the price of explainability for SPEX on k-means experiments?
>
> We include it below. EMN as expected has the lower price of explainability for k-means on most datasets as it is tailored for this reference clustering.
>
> |  | Ecoli | Iris | Cancer | MNIST | Caltech | Newsgroups | Beans | Cifar |
> |---|---|---|---|---|---|---|---|---|
> | SpEx-$k$-means | 1.052 | 1.044 | 1 | 1.176 | 1.595 | 1.209 | 1.005 | 1.259 |
> | EMN-$k$-means | 1.05 | 1.044 | 1 | 1.178 | 1.575 | 1.229 | 1.004 | 1.237 |
> | CART-$k$-means | 1.058 | 1.044 | 1 | 1.512 | 2.588 | 1.29 | 1.004 | 1.419
>
> > Did the authors consider comparing against other practical explainable clustering algorithms [1,2]?
>
> Thank you for the relevant references, we will include a discussion of [1] (while [2] is already referenced as [15]). Both papers use more relaxed notions of explainability than the algorithms we consider: [1] allows more generally shaped clusters (for example, the clustering in Fig. 2(a) in [1] is not expressible in the explainability model from [31]) and [2] extends the decision tree to more than [2] leaves. Nonetheless it is interesting to compare them to SpEx under a proper extension. Though we haven’t found an available implementation of [1], we include below a comparison to ExKMC [2] for explainable clustering with $\ell=2k$ leaves. The results suggest that ExKMC and SpEx-Clique are comparable in this regime, though ExKMC scores higher in ARI while SpEx-Clique scores higher in AMI.
>
> | _ARI_ | Ecoli | Iris | Cancer | MNIST | Caltech-101 | 20Newsgroups | Beans | CIFAR |
> |---|---|---|---|---|---|---|---|---|
> | SpEx-Clique | 0.469 | 0.716 | 0.491 | 0.197 | 0.203 | 0.088 | 0.37 | 0.309 |
> | ExKMC | 0.458 | 0.73 | 0.491 | 0.224 | 0.249 | 0.096 | 0.37 | 0.323 |
> | CART-k-means | 0.447 | 0.716 | 0.491 | 0.031 | 0 | 0.026 | 0.37 | 0.144 |
> | CART-spectral | 0.434 | 0.61 | 0.749 | 0.08 | 0 | 0.006 | 0.608 | 0.135 |
>
> | _AMI_ | Ecoli | Iris | Cancer | MNIST | Caltech-101 | 20Newsgroups | Beans | CIFAR |
> |---|---|---|---|---|---|---|---|---|
> | SpEx-Clique | 0.589 | 0.739 | 0.464 | 0.317 | 0.565 | 0.281 | 0.512 | 0.484 |
> | ExKMC | 0.582 | 0.755 | 0.464 | 0.328 | 0.546 | 0.263 | 0.512 | 0.467 |
> | CART-k-means | 0.573 | 0.739 | 0.464 | 0.177 | 0.208 | 0.148 | 0.512 | 0.371 |
> | CART-spectral | 0.493 | 0.648 | 0.631 | 0.21 | 0.214 | 0.104 | 0.695 | 0.381 |
>
> > Would it not be more reasonable in clustering to colour them according to some $k$-means partition?
>
> This would be reasonable as well, and is what has been used in visualizations in many of the cited prior works. We chose a different notion of reference clustering than $k$-means in our visualization, since for $k$-means near-optimal explainable clustering algorithms were already known, and our focus is on extensions beyond $k$-means to general reference clusterings.

---

> > ### Comment · Reviewer_K9ow · 2025-08-08
> > **Reply**
> >
> > Dear authors, thank you for your answer. I believe that adding the experiments from your rebuttal and a discussion on [1] would help contextualize the contribution of the paper.

---

### Official Review · Reviewer_A43H · 2025-07-02

**Clarity:** 2
**Significance:** 2
**Originality:** 2
**Rating:** 4
**Confidence:** 3

**Summary:**

This paper proposes a spectral graph partitioning-based explainable clustering method to overcome the limitations of previous decision tree-based explainable clustering approaches. It primarily introduces two methods: SPEX-Clique and SPEX-kNN. The paper conducts experiments comparing these methods against EMN, CART, and Kernel IMM, and analyzes the implications of the results.

**Questions:**

- Please clarify the explainability of the proposed model in the context of existing XAI methods.
- The paper primarily evaluates performance using standard clustering metrics (ARI, AMI). While these measure agreement with ground truth, they don't directly assess the quality of the explanation or the interpretability of the generated trees. Please provide a deeper analysis for the explainability.
- Could the authors provide a more robust justification for the greedy approach?

**Ethical Concerns:**

["NO or VERY MINOR ethics concerns only"]

**Final Justification:**

The authors addressed my concerns well about the justification of the greedy approach and time complexity.

**Limitations:**

Yes

**Quality:**

2

**Strengths And Weaknesses:**

## Strengths
- The paper clearly analyzes the limitations of past research and effectively highlights the significant importance of explainability.

- It attempts to overcome the limitations of previous studies by basing its approach on spectral graph partitioning.

- The paper presents experimental results across nine datasets, analyzing the strengths and weaknesses of the proposed methods.


## Weaknesses
- The paper does not sufficiently cover explainability aspects from past research. More recent studies on explainable unsupervised learning could be discussed to provide a broader context.

- The explainability of the proposed model is not clearly understood. The paper's discussion of prior limitations is too broad, making it difficult to clearly discern the specific contributions of this work. For instance, it is challenging to understand how the bound in Theorem 2.2 directly translates to explainable clustering.

- The experimental design and analysis regarding explainability are insufficient. A deeper consideration and analysis of the benefits are needed. Currently, the performance results in Figure 3 show mixed outcomes.

- A detailed time complexity analysis is missing. While the efficiency of storing graphs and the sweep-line procedure are mentioned, a formal analysis of the overall time complexity for the proposed algorithms (SPEX-Clique and SPEX-kNN) would enhance the paper.

- The greedy approach used for building the decision tree could be further justified. A discussion on its potential trade-offs or any theoretical guarantees (even if not strict optimality) would be beneficial.

- The paper mentions that some datasets are "Embedded with CLIP" or "Embedded with SBERT". This crucial information warrants a brief discussion in the experimental setup to explain why these specific embeddings were chosen and their potential impact on the results.

- The impact of the parameter $k$ for SPEX-kNN could be discussed in a more comprehensive manner. While additional results are in Appendix B, a summary of the sensitivity and optimal selection process in the main text would be valuable.

---

> ### Author Rebuttal · Authors · 2025-07-29
>
> Thank you for your thoughtful review and questions.
>
> > proposed model in the context of existing XAI methods, explanation or the interpretability
>
> First, we reiterate that the explainability model we study is not proposed by us, but has been proposed by [31] and widely adopted since (see the references in line 35). Our contribution is in developing new algorithms and theoretical framework for it.
>
> To position their model in XAI: Generally there are two main approaches in XAI for explaining an ML model $M^{\*}$. (i) “Intrinsic explainability”, where $M^{\*}$ is approximated by a different model $M’$ which has a strictly imposed explainable structure. This yields strong interpretability since $M’$ is structurally explainable, while introducing a compromise on accuracy since $M’$ may differ from $M^{\*}$ in its output predictions. (ii) “Post-hoc explainability”, which attempts to match explanations to $M^{\*}$ after the fact while viewing it as an unchanged black-box. This maintains full accuracy since model predictions are never changed, but significantly inhibits interpretability due to the inherent “just so” nature of post-hoc explanations – this approach is bound to force explanations even on unexplainable model outcomes.
>
> The explainable clustering model of [31] belong to type (i) – it is an intrinsically explainable method, where $M^{\*}$ is the reference clustering, $M’$ is the coordinate-threshold decision tree, and the gap in accuracy between them is quantified as the “price of explainability". Explainability is defined as describing the points assigned to each cluster by a sequence of feature thresholds.
>
> > Could the authors provide a more robust justification for the greedy approach?
>
> Prior work has established that greedy algorithms are worst-case near-optimal for explainable $k$-means clustering (in [11, 18]). Worst-case optimality here means that there is an instance where greedy outputs the best possible explainable clustering tree overall for that instance, so no loss is incurred due to the use of a greedy approach over the intrinsic cost of explainability. Together with the obvious efficiency benefits of greedy algorithms, it seems a natural approach for explainable clustering with other reference clusterings as well, even though it remains possible that the greedy approach is not optimal in general.
>
> > how the bound in Theorem 2.2 directly translates to explainable clustering
>
> In summary, the bound shows that there exists a coordinate threshold cut whose conductance (left-hand side) is bounded by the clusterability of the point set (right-hand side). The intrinsic explainability model from [31] defines explainability by coordinate threshold cuts, and the conductance quantifies the loss in accuracy between $M^*$ and $M’$, hence the bound shows the existence of intrinsic explainability with bounded loss in accuracy.
>
> > why these specific embeddings were chosen
>
> CLIP [34] and SentenceBERT [35] are standard, public and widely used methods for generating numerical embeddings for images and for text (respectively), currently used as standard across ML in research papers and applications, and were therefore a natural choice.
>
> > the parameter $k$ for SPEX-kNN
>
> The $k$ parameter in kNN graph construction is a long studied topic in ML. Very small $k$ may lead to bad or disconnected graphs due to too few edges, while very large $k$ may lead to connecting faraway and dissimilar points, and therefore intermediate values are recommended. The common guideline is to test values in the range 10-20 empirically with the “elbow method” that selects the value where performance plateaus before degrading. This is the procedure we followed except that we chose a single value across all datasets for methodological uniformity and soundness.
>
> > detailed time complexity analysis
>
> Please see the response to eAyA for the time complexity analysis.

---

> > ### Comment · Reviewer_A43H · 2025-08-08
> >
> > The authors addressed my concerns well, and I have increased my score accordingly. Thank you for all your efforts.

---

### Official Review · Reviewer_eAyA · 2025-07-04

**Clarity:** 4
**Significance:** 4
**Originality:** 3
**Rating:** 5
**Confidence:** 4

**Summary:**

The paper introduces SpEx, a method to explain clusters of points in $\mathbb{R}^d$ using decision trees. SpEx is part of the literature that aims to explain clusters of points using binary decision trees. The main advantage of SpEx is its generality in handling input clusters. While previous methods mostly work for centroid-based clustering methods (e.g., k-means and k-median), SpEx aims to generate binary decision tree-based explanations for generic input clusters. Based on a theoretical observation, the paper proposes two instantiations of SpEx (SpEx-Clique and SpEx-kNN). The paper also performs experimental evaluations of their method against other baselines on a range of datasets.

**Questions:**

* How sensitive is SpEx’s performance to the choice of the number of leaves $\ell$? Do the authors have any principled guidelines (empirical or theoretical) for selecting $\ell$ beyond simply matching the number of clusters?

* What happens if I decide to prune the trees after finding them? Pruning may help with the interpretability of the tree.

* Can you provide more insight or analysis into the observed gap between low- and high-dimensional datasets? In particular, is this some sort of “curse of dimensionality” effect?

* Could you bring Tab. 1 and Tab. 2 into the main text paper?

* Could you add further discussion on the complexity of the algorithms? Perhaps list the theoretical (experimental) complexities in a comparison table?

**Ethical Concerns:**

["NO or VERY MINOR ethics concerns only"]

**Final Justification:**

I recommend the paper be accepted.

The problem solved by the authors is of interest to part of the interpretability community, especially the sub-group of researchers that are interested in "white-box" model explanations for clustering.

In addition, my comments and those of most other reviewers are minor changes in the paper that mainly improve the clarity of the presented results and enhance the presentation of the main contributions.

Finally, I believe that this paper will be appreciated by the NeurIPS interpretability community and may have a significant impact after the authors implement the changes we (all reviewers) suggested.

**Limitations:**

The authors discuss the limitations of their approach. My only suggestion is the addition of a discussion on the interpretability of decision trees with a large number of leaves (e.g., 100 leaves). Are large decision trees really interpretable?

**Paper Formatting Concerns:**

* You may want to start sentences with \citet instead of cite.

* You are inconsistently using eq. and Equation. Check your text and use only one of them. Specifically, when referencing an equation, it may be better to just use \eqref and not even write eq. nor Equation. Check lines 181, for example.

* In line 346 you have two commas. "3.3,,".

**Quality:**

3

**Strengths And Weaknesses:**

### Strengths

* The paper proposes a solution to the interesting problem of explaining clusters using decision trees without strong assumptions about the cluster structure.

* The authors propose a theoretical framework to describe other explainability methods through a similar lens as SpEx. This is an interesting contribution to this literature.

* The paper makes a thorough performance comparison across datasets, but some of the results are in the appendix.

* The paper is well organized and does a good job of introducing the problem of cluster explainability to unfamiliar readers.

### Weaknesses

* The provided instantiations of SpEx don’t have strong performance in scenarios where other methods perform well. This is expected, since SpEx makes fewer assumptions about the problem’s structure, hence not a major weakness.

* There is no discussion on how to select the number of leaves. The proposed algorithms introduce one more parameter, the number of leaves $\ell$, which the authors set to the number of clusters. Although other methods in the literature employ a similar choice, the generality of SpEx raises questions about whether this is the optimal choice for the number of leaves. The paper presents no experimental or theoretical evidence to support their design choice.

* Important experiments only appear in the appendix (Tab. 2 and Tab. 3). These are important observations, which are relevant to the main conclusion of the paper, and hence should be brought to the main paper.

* There seems to be a gap in performance in low and high-dimensional input spaces. The authors do not discuss this interesting finding enough in the paper. Is there an explanation for this finding on the style of “the curse of dimensionality”?

---

> ### Author Rebuttal · Authors · 2025-07-29
>
> Thank you for your valuable review and support!
>
> > Can you provide more insight or analysis into the observed gap between low- and high-dimensional datasets?
>
> The main source of the gap is the axis-aligned nature of explainable clustering: the number of coordinates that the decision tree reads from every point is at most the depth of the tree, hence if the number of clusters (and thus the depth of the tree) is much smaller than the dimension, the tree has to make clustering decisions based on only a small fraction of coordinates. This renders high-dimensional datasets more difficult to cluster with explainability.
>
> > Could you bring Tab. 1 and Tab. 2 into the main text paper?
>
> Yes, we will add it.
>
> > Could you add further discussion on the complexity of the algorithms?
>
> SpEx as well as the baselines share the following high-level structure. In each tree node $u$ with $n_u$ points, for each coordinate, they sort the points by that coordinate (time $O(n_u\log n_u)$) and compute a score for each of the $n_u-1$ possible prefix/suffix cuts (time $O(n_uS)$, where $S$ is the time to compute a cut score). Repeating this for all coordinates takes time $O(dn_u(\log n_u + S))$. In each tree level the sum of $n_u$s over the nodes $u$ in that level is $n$ for SpEx and CART, and $n+k\leq 2n$ for IMM and EMN, so the time per level in all of them is $O(dn(\log n + S))$. Summing over up to $k-1$ levels in the tree, the total time is $O(kdn(\log n + S))$.
>
> Where the algorithms may differ is in the score they use and the time $S$ it takes to compute it. This requires analyzing each algorithm separately, since they use different scores, but it turns out that in all of them (IMM, EMN, CART, SpEx) the score can be computed in time $O(1)$ with a sweep-line procedure (more precisely, the score can be updated in time $O(1)$ as we iterate over prefix/suffix cuts). In the graph-theoretic lens from section 3, whether the score admits an efficient sweep-line procedure depends on the graphs $G,H$: generally if they are highly structured (like the clique, star, IS, etc) then such procedure is possible. Thus all algorithms have the same asymptotic time complexity $O(kdn\log n)$.
>
> > choice of the number of leaves $\ell$
>
> > discussion on the interpretability of decision trees with a large number of leaves (e.g., 100 leaves). Are large decision trees really interpretable?
>
> These are important inter-connected points. The setting of $\ell$ relative to the number of reference clusters $k$ is a use-case dependent choice that governs how much the user prioritizes explanability versus accuracy. Viewing clustering as an unsupervised ML model, the number of reference clusters $k$ is the number of possible model outcomes, and the number of leaves $\ell$ in the explainable decision tree is the number of explanations. Since each outcome must have its own explanation, we must have $\ell\geq k$. The case $k=\ell$ is the most explainable since there is one explanation per outcome – e.g., “age in [range] and zip code in [range]” for assignments to public schools. As $\ell$ is increased, accuracy improves since a bigger tree is better able to approximate the reference clustering, however explainability degrades as now the same model outcome would have multiple different explanations (e.g., “age in [range] and zip code in [range] OR age in [range] and GPA above [bar]”). In the limit each point may have its own separate explanation, which is in a sense no explanation at all.
>
> Thus, we agree that if the number of leaves is much larger than the number of reference clusters, the decision tree can no longer be considered truly explainable. This is why we focused on the most explainable case, $\ell=k$. In the response to K9ow we include additional experiments with $\ell=2k$ for reference.
>
> > What happens if I decide to prune the trees after finding them?
>
> Since pruning decreases the number of leaves, it could help in pruning a tree constructed with $\ell>k$ leaves back to $k$ leaves (or to an intermediate value). Since pruning has the “benefit of hindsight” over greedy partitioning, it could potentially lead to better trees, depending on the pruning policy. This is an interesting direction to explore.

---

> > ### Comment · Reviewer_eAyA · 2025-08-08
> >
> > I thank the authors for their response and efforts during the rebuttal.
> > All my concerns were addressed and I am still positive about the paper.
> >
> > I have also read the comments from the other reviewers and:
> >
> > 1. I agree with the reviewer K9ow that the additional experiments would help clarify the main contributions of the paper.
> >
> > 2. After reading the reviewer cfLS comments, I also believe the paper may benefit from a more thorough literature review as suggested.
> >
> > I hope the authors can make use of the camera-ready paper to address our comments.

---

### Note · Authors · 2025-08-13

We thank all the reviewers for their in-depth reading of our paper and their supportive and valuable feedback. During the discussion phase we have clarified and elaborated on points inquired by the reviewers and included complementary experimental results to support the discussion. We summarize our high-level takeaways:

* Presentation: elaborate on the explainability aspect of the model, particularly the wider context in XAI and the relation between the number of clusters and number of leaves. Include the detailed runtime analysis.

* Related work: extend the literature review to encompass the additional references raised during the discussion.

* Experiments: Include the tabular experimental results from the appendix to the main text, incorporate additional experimental results from the discussion phase.

All specific comments would be addressed as well.

---

### Decision · Program_Chairs · 2025-09-17

**Decision:**

Accept (poster)

**Comment:**

This paper makes a nice contribution to the literature of explainable clustering by tying explainability to natural structures in an associated graph (existing explainability framework actually considers decision tree representation). The vertices of the graph are identified with the d-dimensional data points. The main tool is a theorem that shows that the conductance of a coordinate cut (subset of vertices with a specific coordinate smaller than some value) is upper bounded by a quantity "proportional" to some "average distance" between adjacent vertices of the graph. From such a graph then a decision tree is constructed where every internal node is associated with a coordinate and threshold (thus guaranteeing explainability in the usual sense).

This is a novel approach to explainable clustering that is useful and interesting in my reading.

The reviews are positive about the contribution; the reviewers are also satisfied with the comments made during the rebuttal phase. I recommend acceptance.